# Are Your Models Still Fair? Fairness Attacks on Graph Neural Networks via Node Injections

**Zihan Luo, Hong Huang**[†]**, Yongkang Zhou, Jiping Zhang, Nuo Chen, Hai Jin**
Huazhong University of Science and Technology, China
{`zihanluo,honghuang,yongkangzhou_,jipingz,nuo_chen,hjin`}@hust.edu.cn

## Abstract

Despite the remarkable capabilities demonstrated by *Graph Neural Networks* (GNNs) in graph-related tasks, recent research has revealed the fairness vulnerabilities in GNNs when facing malicious adversarial attacks. However, all existing fairness attacks require manipulating the connectivity between existing nodes, which may be prohibited in reality. To this end, we introduce a *Node Injection-based Fairness Attack* (NIFA), exploring the vulnerabilities of GNN fairness in such a more realistic setting. In detail, NIFA first designs two insightful principles for node injection operations, namely the uncertainty-maximization principle and homophily-increase principle, and then optimizes injected nodes' feature matrix to further ensure the effectiveness of fairness attacks. Comprehensive experiments on three real-world datasets consistently demonstrate that NIFA can significantly undermine the fairness of mainstream GNNs, even including fairness-aware GNNs, by injecting merely 1% of nodes. We sincerely hope that our work can stimulate increasing attention from researchers on the vulnerability of GNN fairness, and encourage the development of corresponding defense mechanisms. Our code and data are released at: https://github.com/CGCL-codes/NIFA.

## 1 Introduction

Due to the strong capability in understanding graph structure, *Graph Neural Networks* (GNNs) have achieved much progress in graph-related domains such as social recommendation [30, 31] and bioinformatics [4, 27]. Nevertheless, despite the impressive capabilities demonstrated by GNNs, more and more in-depth research has revealed shortcomings in the fairness of GNN models, which greatly restricts their applications in the real world.

In fact, studies [5, 42] have found that the biases and prejudices existed in training data would be further amplified through the message propagation mechanism of GNNs, leading to model predictions being correlated with certain sensitive attributes, such as *gender* and *race*. Such correlations are usually undesired and can result in fairness issues and societal harm. For instance, in online recruitment, a recommender based on GNNs may be associated with the gender of applicants, leading to differential treatments towards different demographics and consequently giving rise to group unfairness. To address fairness issues in GNNs, researchers have proposed solutions such as adversarial learning [5, 28], data augmentation [17, 22] and others, which have achieved promising results.

However, recent research in the machine learning domain indicates that fairness is actually susceptible to adversarial attacks [3, 26, 29]. Given this, we cannot help but wonder: *"Is the fairness of GNN*

---

[†]Hong Huang is the corresponding author. Zihan Luo, Hong Huang, Yongkang Zhou, Jiping Zhang and Hai Jin are affiliated with the National Engineering Research Center for Big Data Technology and System, Service Computing Technology and Systems Laboratory, Cluster and Grid Computing Lab, School of Computer Science and Technology, Huazhong University of Science and Technology.

38th Conference on Neural Information Processing Systems (NeurIPS 2024).

*models also highly vulnerable?"* For example, in e-commerce, if attackers could exacerbate performance disparities between male and female user groups by attacking GNN-based recommendation models, they could ultimately cause the e-commerce platform to provoke dissatisfaction from specific user demographics and gradually lose its appeal among these users. Several studies [11, 13, 41] have explored the vulnerability of GNN fairness and proposed effective attack strategies. Unlike conventional attacks, these fairness attacks aim to undermine GNN fairness without excessively compromising its utility. However, all these works require altering connectivity between existing nodes, whose authority is typically limited in the real world, such as modifying the relationship between real users. In contrast, injecting fake nodes into the original graph is a more practical way to launch an attack without manipulating the existing graph [12, 32, 34], which is still under-explored in the field of GNN fairness attack. To address this gap, we aim to be the first to launch an attack on GNN fairness via node injection, examining their vulnerabilities under such a more realistic setting.

Specifically, launching a node injection-based fairness attack on GNNs is non-trivial, whose challenges can be summarized as follows: **RQ1:** *How to determine the node injection strategy?* The node injection can be decomposed into two steps, including selecting appropriate target nodes and connecting the injected nodes with them, both will impact the effectiveness of the attack. **RQ2:** *How to determine the features of the injected nodes after node injection?* Like the real nodes, injected nodes will also participate in the message propagation process of GNNs, thereby affecting their neighbors and even the whole graph. Given the key role of massage propagation in GNN fairness [40, 45], proposing suitable strategies to determine the features of injected nodes is also important.

To address these challenges, we propose a gray-box poisoning attack method namely *Node Injection-based Fairness Attack* (NIFA) during the GNN training phase. In detail, for the two steps in the first challenge, NIFA innovatively designs two corresponding principles. The first is the *uncertainty-maximization principle*, which asks to select real nodes with the highest model uncertainty as target nodes for injection. The idea is that nodes with higher uncertainty are typically more susceptible to attacks, thereby ensuring the attack's effectiveness. After selecting target nodes, NIFA follows the second principle, the *homophily-increase principle*, to connect target nodes with injected nodes. This principle aims to deteriorate GNN fairness by enhancing message propagation within sensitive groups [22, 40]. For the second challenge, multiple novel objective functions are proposed after node injection to guide the optimization of the injected nodes' features, which could further impact the victim GNN's fairness from a feature perspective. In summary, our contributions are as follows:

- To the best of our knowledge, we are the first to conduct fairness attacks on GNNs via node injections, and our work successfully highlights the vulnerability of GNN fairness. We also summarize several key insights for the future defense of GNNs' fairness attacks from the success of NIFA.

- We propose a node injection-based gray-box attack named NIFA. To be concrete, NIFA first designs two novel principles to guide the node injection operations from a structure perspective, and then proposes multiple objective functions for the injected nodes' feature optimization.

- We conduct extensive experiments on three real-world datasets, which consistently show that NIFA can effectively attack existing GNN models with only a 1% perturbation rate and an unnoticeable utility compromise, even including fairness-aware GNN models. Comparisons with other state-of-the-art baselines also verify the superiority of NIFA in achieving fairness attacks.

## 2  Related work

**Fairness on GNNs.** Researchers have discovered various fairness issues of GNNs, which often lead to societal harms [22, 37] and performance deterioration [21, 33] in practical applications. Algorithmic fairness on GNNs can be categorized into two main types based on the definition: individual fairness [1, 6] and group fairness [5, 37, 46]. Individual fairness requires that similar individuals should receive similar treatment, while group fairness aims to protect specific disadvantaged groups [16]. In detail, many researchers have delved into studies focusing on fairness grounded in sensitive attributes. For instance, Dai et al. [5] reduce the identifiability of sensitive attributes in node embeddings through adversarial training to enhance fairness. FairVGNN [38] goes a step further by introducing a feature masking strategy to address the problem of sensitive information leakage during the feature propagation process in GNNs. Graphair [17] achieves fairness through an automated data augmentation method and FairSIN [40] designs a novel sensitive information neutralization method for fairness. Beyond fairness related to sensitive attributes, some researchers also direct

attention to fairness related to graph structures, like DegFairGNN [21] and Ada-GNN [23]. In this work, we mainly focus on attacks on the group fairness of GNNs based on sensitive attributes.

**Attacks on GNNs.** Finding out potential vulnerabilities thus improving the security of GNNs remains a pivotal concern in the field of trustworthy GNNs [42]. From the perspective of attackers, they aim to compromise the GNNs' performance on graph data via manipulating graph structures [24, 32, 47], node attributes [48], or node labels [19]. Among these methods, node injection attacks, given the attackers' limited authority to manipulate the connectivity between existing nodes, emerge as one of the most prevalent methods [32, 34, 43]. However, existing attacks, including node injection attacks, mainly focus on undermining GNN's utility, with little attention to the vulnerability of GNN fairness. Different from attacks on GNN utility, fairness-targeted attacks aim to deteriorate the fairness without significantly compromising the accuracy. FA-GNN [11], FATE [13], and G-FairAttack [41] stand out as the few ones that we are aware of to explore attacks on GNN fairness. FA-GNN's empirical findings suggest that adding edges with certain strategies can significantly compromise GNN fairness without affecting its performance [11]. FATE [13] formulates the fairness attacks as a bi-level optimization problem and proposes a meta-learning-based framework. G-FairAttack [41] designs a novel surrogate loss with utility constraints to launch the attacks in a non-gradient manner. Nevertheless, all these works require modifying the link structure between existing nodes, which may be prohibited in reality due to the lack of authority.

# 3 Preliminary

Here we will introduce some basic notations and concepts, and then give our problem definition.

## 3.1 Notations

A graph is denoted as $\mathcal{G} = (\mathcal{V}, \mathbf{A}, \mathbf{X})$, where $\mathcal{V}$ is the node set, and $\mathbf{A} \in \mathbb{R}^{|\mathcal{V}| \times |\mathcal{V}|}$ represents the adjacency matrix. $\mathbf{X} \in \mathbb{R}^{|\mathcal{V}| \times D}$ denotes the feature matrix, in which $D$ is the feature dimension. Under the settings of node classification, each node $v \in \mathcal{V}$ will be assigned with a label $y_v \in \mathcal{Y}$, and a GNN-based mapping function $f_\theta : \{\mathcal{V}, \mathcal{G}\} \rightarrow \{1, 2, ..., |\mathcal{Y}|\}^{|\mathcal{V}|}$ with parameters $\theta$ is learned to leverage the graph signals for label prediction, where $\mathcal{Y}$ represents the true label set.

## 3.2 Fairness-related concepts

In alignment with prior works [5, 7, 17], we mainly focus on group fairness where each node will be assigned with a binary sensitive attribute $s \in \{0, 1\}$, although our attack could also be generalized to the settings of multi-sensitive groups and we leave this as our future work. Based on the sensitive attributes, the nodes can be divided into two non-overlapped groups $\mathcal{V} = \{\mathcal{V}_0, \mathcal{V}_1\}$, and we employ the following two kinds of fairness related definitions:

**Definition 1.** *Statistical Parity (SP). The Statistical Parity requires the prediction probability distribution to be independent of sensitive attributes, i.e. for any class $y \in \mathcal{Y}$ and any node $v \in \mathcal{V}$:*

$$P(\hat{y}_v = y|s = 0) = P(\hat{y}_v = y|s = 1), \tag{1}$$

*where $\hat{y}_v$ denotes the predicted label of node $v$.*

**Definition 2.** *Equal Opportunity (EO). The Equal Opportunity requires that the probability of predicting correctly is independent of sensitive attributes, i.e. for any class $y \in \mathcal{Y}$ and any node $v \in \mathcal{V}$, we can have:*

$$P(\hat{y}_v = y|y_v = y, s = 0) = P(\hat{y}_v = y|y_v = y, s = 1). \tag{2}$$

Based on the above definitions, we can define two kinds of metrics $\Delta_{SP}$ and $\Delta_{EO}$ to quantitatively measure fairness. For both metrics, smaller values indicate better fairness:

$$\Delta_{SP} = \mathbb{E}|P(\hat{y} = y|s = 0) - P(\hat{y} = y|s = 1)|, \tag{3}$$

$$\Delta_{EO} = \mathbb{E}|P(\hat{y} = y|y = y, s = 0) - P(\hat{y} = y|y = y, s = 1)|. \tag{4}$$

### 3.3 Problem definition

In this paper, our goal is to launch fairness-targeted attacks on GNN models through the application of node injection during the training phase, i.e. poisoning attack. Following the line of previous attacks on GNNs [32, 41], our attack is under the prevalent gray-box setting, where the attackers can obtain the graph $\mathcal{G}$ with node labels $\mathcal{Y}$, and the sensitive information $s$, but can not access the model architecture and parameters $\theta$. Detailed introduction to our attack settings is provided in Appendix B. Specifically, through injecting malicious node set $\mathcal{V}_I$ into the graph, the original graph $\mathcal{G} = (\mathcal{V}, \mathbf{A}, \mathbf{X})$ is poisoned as $\mathcal{G}' = (\mathcal{V}', \mathbf{A}', \mathbf{X}')$, where

$$\mathcal{V}' = \mathcal{V} \cup \mathcal{V}_I, \ \mathbf{X}' = \left[ \begin{array}{c} \mathbf{X} \\ \mathbf{X}_I \end{array} \right], \mathbf{X}_I \in \mathbb{R}^{|\mathcal{V}_I| \times D}, \tag{5}$$

$$\mathbf{A}' = \left[ \begin{array}{cc} \mathbf{A} & \mathbf{V}_I \\ \mathbf{V}_I^T & \mathbf{A}_I \end{array} \right], \mathbf{V}_I \in \mathbb{R}^{|\mathcal{V}| \times |\mathcal{V}_I|}, \mathbf{A}_I \in \mathbb{R}^{|\mathcal{V}_I| \times |\mathcal{V}_I|}. \tag{6}$$

Both $\mathbf{V}_I$ and $\mathbf{A}_I$ are matrices for illustrating the connectivity related to injected nodes, and $\mathbf{X}_I$ is the feature matrix of injected nodes $\mathcal{V}_I$. The true label set $\mathcal{Y}$ will not be poisoned by injected nodes in our settings, as such information is typically hard to modify in reality. For conciseness, we denote $\mathcal{F}(\cdot)$ and $\mathcal{M}(\cdot)$ as the evaluation functions on fairness and utility for the learned mapping function $f_\theta$, respectively. Then our goal as an injection-based attack on fairness could be formulated as:

$$\max_{\mathcal{G}'} |\mathcal{F}(f_{\theta^*}(\mathcal{V}, \mathcal{G}'))| \tag{7}$$

$$\text{s.t. } \arg\max_{\theta^*} \mathcal{M}(f_{\theta^*}(\mathcal{V}, \mathcal{G}')), \ \mathcal{G}' = (\mathcal{V}', \mathbf{A}', \mathbf{X}'), \ |\mathcal{V}_I| \le b, \ deg(v)_{v \in \mathcal{V}_I} \le d.$$

As a poisoning attack, the first constraint in Eq. (7) requires to train the victim model $f_{\theta^*}$ with parameters $\theta^*$ on the poisoned graph $\mathcal{G}'$, so that the predictions of $f_{\theta^*}$ are as correct as possible before evaluating the attack performance. The following constraints in Eq. (7) make sure that the proposed attack is unnoticeable and deceptive to the defenders, i.e. the number of injected nodes is below a predefined budget $b$[1] and the degrees of injected nodes are constrained by a budget $d$. Our goal is to find a poisoned graph $\mathcal{G}'$ to deteriorate the fairness of victim models $f_{\theta^*}$ as severely as possible, i.e. maximize the fairness metrics $\Delta_{SP}$ and $\Delta_{EO}$ introduced previously.

## 4 Methodology

In this section, we first give an overview of our attack method NIFA. Then we will elaborate on the details of each module and summarize the implementation algorithms at last.

### 4.1 Framework overview

The overall framework of NIFA is illustrated in Figure 1. As mentioned before, NIFA first employs two principles to guide the node injection operations. For the first uncertainty-maximization principle, NIFA utilizes the Bayesian GNN for model uncertainty estimation of each node, and then selects target nodes with the highest uncertainty (a). As for the second homophily-increase principle, NIFA requires each injected node can only establish connections to target nodes from one single sensitive group (b), thus increasing the homophily-ratio and enhancing information propagation within sensitive groups. After node injection, multiple objective functions are designed to guide the optimization of injected nodes' feature matrix, where we introduce an iterative optimization strategy for avoiding over-fitting issues (c). The details of each part will be introduced later.

### 4.2 Node injection with principles

The first step of NIFA is conducting node injections, which aims to ensure the effectiveness of NIFA from a structure perspective. In detail, we propose two novel principles to guide the node injection operations, namely *Uncertainty-maximization principle* and *Homophily-increase principle*.

---

[1]Same as the prior work [32], we define the perturbation rate as the ratio of injected nodes to the labeled nodes in the original graph, i.e. $|\mathcal{V}_I|/|\mathcal{V}_L|$, where $\mathcal{V}_L$ denotes the labeled node set.

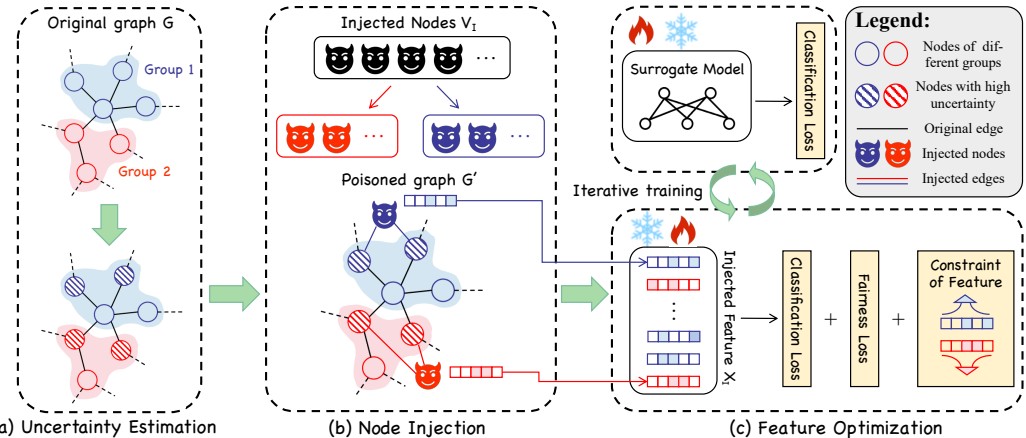

Figure 1: The overall framework of NIFA: (a) Utilizing uncertainty estimation, nodes exhibiting high uncertainty (depicted as shaded nodes) are designated as targeted nodes. (b) Injected nodes are equally assigned to each sensitive group, and only connect targeted nodes with the same sensitive attribute. (c) After node injection, the injected feature matrix and surrogate model are optimized iteratively by diverse objective functions.

**Uncertainty-maximization principle.** Intuitively, nodes with higher model uncertainties are positioned closer to the decision boundary, which means their predicted labels are more vulnerable and easier to flip when facing adversarial attacks. We acknowledge that the model uncertainty may not be the only method to measure the vulnerability of nodes, and we will discuss potential alternative approaches in Appendix H.2. Inspired by [20], we utilize a Bayesian GNN to estimate the model uncertainty of each node, where we employ the Monte Carlo dropout approach [9] to approximate the distributions of the sampled model parameters. Given a GNN with parameters $\theta_\mathcal{B}$, we obtain different model parameters through $T$ times independent Bernoulli dropout sampling processes, i.e.:

$$\begin{aligned} P(M_i) &\sim \text{Bernoulli}(p) \\ \theta_{\mathcal{B}_i} &= M_i \odot \theta_\mathcal{B} \end{aligned}, i \in \{1, 2, \ldots, T\}, \tag{8}$$

where $M_i$ is the $i$th sampled binary mask following the Bernoulli distribution with parameter $p$, and $\odot$ denotes the dot production operation. Here we take a two-layer GCN [10] with parameters $\theta_\mathcal{B}$ as the Bayesian GNN for estimating the uncertainty of each node, and $\theta_\mathcal{B}$ is optimized by minimizing the following objective function, which consists of a cross-entropy loss plus a regularization term:

$$L(\theta_\mathcal{B}) = -\frac{1}{T} \sum_{i=1}^{T} \mathcal{Y} \log(f_{\theta_{\mathcal{B}_i}}(\mathcal{V}, \mathcal{G})) + \frac{1-p}{2T} \|\theta_\mathcal{B}\|_2^2, \tag{9}$$

where $\mathcal{Y}$ denotes the true labels, $\|\cdot\|_2^2$ denotes the L2 regularization, $T$ is the number of sampling processes and $f_{\theta_{\mathcal{B}_i}}(\cdot)$ is the mapping function with the $i$th sampled parameters $\theta_{\mathcal{B}_i}$. After the training process, model uncertainty can be estimated by calculating the variance of $T$ times predictions with the sampled parameters $\{\theta_{\mathcal{B}_i}\}_{i=1}^{T}$. Intuitively, nodes with lower variance are more confident in their predictions and vice versa. Thus, the model uncertainty scores $U \in \mathbb{R}^{|\mathcal{V}|}$ are positively correlated with the model prediction variance, and we simply estimate $U$ with the following formulation:

$$U = \underset{i=1}{\overset{T}{Var}}(f_{\theta_{\mathcal{B}_i}}(\mathcal{V}, \mathcal{G})). \tag{10}$$

Under the guidance of the uncertainty-maximization principle, we will select nodes with the top $k\%$ model uncertainty $U$ in each sensitive group as the target nodes, where $k$ is a hyper-parameter.

**Homophily-increase principle.** After selecting target nodes, the next step is to connect the injected nodes with them. In particular, we first present our strategy in this step, with more rationales provided later: the injected nodes $\mathcal{V}_I$ are first equally assigned to each sensitive group in the graph, then each injected node will exclusively connect to $d$ random target nodes with the same sensitive attribute, as illustrated in Figure 1(b), where $d$ is a hyper-parameter. At this stage, the node injection operations are completed, with the structure of the original graph $\mathcal{G}$ manipulated.

Intuitively, compared with random node injection, our strategy prevents information propagation between nodes of different sensitive groups through the injected nodes, making it easier to accentuate

differences in embeddings between groups and thereby exacerbate unfairness issues [37]. We also provide a brief theoretical analysis to show that such a strategy could lead to the increase of node-level homophily ratio. Similar to [8], we define the node-level homophily-ratio $\mathcal{H}_u$ as the ratio of neighbors of node $u$ that have the same sensitive attribute as node $u$, i.e. $\mathcal{H}_u = \frac{\sum_{v \in \mathcal{N}_u} \mathbb{1}(s_u = s_v)}{|\mathcal{N}_u|}$, where $\mathcal{N}_u$ denotes the neighbors of node $u$ and $\mathbb{1}(\cdot)$ is an indicator function. Then we can have:

**Lemma 1.** *For target node $u$ that will connect with injected nodes, our proposed node injection strategy will lead to the increase of node-level homophily-ratio $\mathcal{H}_u$.*

Due to space limitation, the proof for Lemma 1 is provided in Appendix C. It is worth noting that $\mathcal{H}_u$ is also equivalent to the probability of choosing neighbors with the same sensitive attribute for node $u$. From the perspective of message propagation, higher node-level homophily-ratio indicates that more sensitive-related information will be aggregated to the target node, thus leading to more severe unfairness issues on sensitive attributes[2]. We believe that such characteristics could empower our node injection strategy with stronger capability on fairness attacks.

### 4.3 Feature optimization

In this part, we will introduce the details of optimizing injected nodes' features $\mathbf{X}_I$, which helps further advance the effectiveness of NIFA. Generally, under a gray-box attack setting, there is no visible information about the victim models for the attackers, thus requiring attackers to propose a surrogate GNN model $\mathcal{S}$ at first for assessing their attacks. To be specific, similar to the training process of victim models as described in Eq. (7), the surrogate model $\mathcal{S}$ will be trained on the poisoned graph $\mathcal{G}'$ and optimize its parameter $\theta_\mathcal{S}$ to maximize the utility. Conversely, $\mathbf{X}_I$ is designed to mislead $\mathcal{S}$, ensuring that even a well-trained surrogate model will still maintain high unfairness under attacks. Instead of employing a pre-trained frozen surrogate model $\mathcal{S}$, NIFA asks two components, i.e. $\mathcal{S}$ and $\mathbf{X}_I$ to be trained iteratively with different objective functions, which avoids the attack being over-fitting to specific model parameters. In detail, the surrogate model $\mathcal{S}$ follows the common training procedure of a GNN classifier with cross-entropy loss, while for the injected nodes' feature optimization, we devise multiple effective objective functions as follows:

**Classification loss.** Although our primary goal is to maximize the unfairness of a GNN model, it is crucial to ensure that the utility of the victim model will not experience a significant decrease after training on a poisoned graph [11, 13, 41], thus being unnoticeable for utility-based attack detection. To this end, we set cross-entropy loss as our first objective function, i.e.:

$$L_{CE} = -\frac{1}{|\mathcal{V}^{tr}|} \sum_{u \in \mathcal{V}^{tr}} y_u \log h_u, \tag{11}$$

where $\mathcal{V}^{tr}$ denotes the original training node set, and $h_u$ denotes the output logits of node $u$.

**Fairness loss.** Aiming at enlarging the unfairness on GNNs, we then design two kinds of fairness loss based on the definitions of $\Delta_{SP}$ and $\Delta_{EO}$, which are formulated as:

$$L_{SP} = -\| \frac{1}{|\mathcal{V}_0^{tr}|} \sum_{u \in \mathcal{V}_0^{tr}} h_u - \frac{1}{|\mathcal{V}_1^{tr}|} \sum_{u \in \mathcal{V}_1^{tr}} h_u \|_2^2, \tag{12}$$

$$L_{EO} = -\| \sum_{y \in \mathcal{Y}} ( \frac{1}{|\mathcal{V}_{0,y}^{tr}|} \sum_{u \in \mathcal{V}_{0,y}^{tr}} h_{u,y} - \frac{1}{|\mathcal{V}_{1,y}^{tr}|} \sum_{u \in \mathcal{V}_{1,y}^{tr}} h_{u,y} ) \|_2^2, \tag{13}$$

where $h_u \in \mathbb{R}^{|\mathcal{Y}|}$ denotes the raw output of node $u$, and $h_{u,y} \in \mathbb{R}$ denotes the raw output of node $u$ on class $y$. $\mathcal{V}_{i,y}^{tr}$ denotes the training nodes with sensitive attribute $i$ and label $y$. By minimizing $L_{SP}$ and $L_{EO}$, the gap in output between different groups increases, leading to high unfairness.

**Constraint of feature.** To further accentuate the differences between different sensitive groups, it is important to ensure that the information introduced by injected nodes for different sensitive groups

---

[2]Similar conclusions have been concluded from multiple prior works [22, 26, 40]. For better a understanding of the relationship between the homophily-ratio and unfairness, one can also refer to [40], which provides the corresponding theoretical analysis from a massage propagation perspective.

is distinct during the message propagation process. To this end, we devise the following constraint function on the injected node feature matrix $\mathbf{X}_I$:

$$L_{CF} = -\left\| \frac{1}{|\mathcal{V}_{I,0}|} \sum_{u \in \mathcal{V}_{I,0}} \mathbf{X}_{I,u} - \frac{1}{|\mathcal{V}_{I,1}|} \sum_{u \in \mathcal{V}_{I,1}} \mathbf{X}_{I,u} \right\|_2^2, \tag{14}$$

where $\mathcal{V}_{I,i}$ is the injected node set linking to the $i$th sensitive group during the node injection.

**Overall loss.** By combining the aforementioned objective terms, the overall loss $L$ for injected nodes' features optimization can be formulated as:

$$L = L_{CE} + \alpha \cdot L_{CF} + \beta \cdot (L_{SP} + L_{EO}), \tag{15}$$

where $\alpha$ and $\beta$ are two hyper-parameters to control the weights of different objective functions.

### 4.4 Implementation algorithm

**Training process.** Due to space limitation, we summarize the pseudo-code of NIFA in Algorithm 1 in Appendix D. Initially, we perform node injection operations based on two proposed principles (lines 2-4). Subsequently, an iterative training strategy is utilized to optimize the surrogate model and injected nodes' feature (lines 5-15). Specifically, after each inner loop for $\mathbf{X}_I$ training, it is clamped to fit the range of the original feature $\mathbf{X}$ (line 14) so that the defenders cannot filter out the injected nodes easily through abnormal feature detection. For datasets with discrete features, $\mathbf{X}_I$ is rounded to the nearest integer at the end of the training process (lines 16-18).

**Inference process.** As a poisoning attack, the original clean graph $\mathcal{G}$ is poisoned as $\mathcal{G}'$ after malicious node injection and feature optimization. The victim models will re-train on the poisoned graph $\mathcal{G}'$ normally, and we take the predictions from the poisoned victim model for final evaluation.

## 5 Experiments

### 5.1 Experimental settings

**Datasets.** Experiments are conducted on three real-world datasets namely Pokec-z, Pokec-n, and DBLP. Both **Pokec-z** and **Pokec-n** are subgraphs sampled from Pokec, one of the largest online social networks in Slovakia, according to the provinces of users [5]. Each node in these graphs represents a user, while each edge represents an unidirectional following relationship. The datasets provide node attributes including age, gender, and hobbies, and the classification task is to predict the working fields of users. **DBLP** is a coauthor network dataset [11], where each node represents an author and two authors will be connected if they publish at least one paper together. The node features are constructed based on the words selected from the corresponding author's published papers. The final classification task is to predict the research area of the authors. The detailed dataset statistics are summarized in Table 1.

Table 1: Dataset statistics

| Dataset | Pokec-z | Pokec-n | DBLP |
|---|---|---|---|
| # of nodes | 67,796 | 66,569 | 20,111 |
| # of edges | 617,958 | 517,047 | 57,508 |
| feature dim. | 276 | 265 | 2,530 |
| # of labeled nodes | 10,262 | 8,797 | 3,196 |

**Victim models.** As a gray-box attack method, we target multiple classical GNNs as victim models, including GCN [14], GraphSAGE [10], APPNP [15], and SGC [39]. We also include three well-established fairness-aware GNNs — FairGNN [5], FairVGNN [38], and FairSIN [40] as our selected victim models. The details of these victim models will be elaborated in Appendix E.

**Baselines.** Depending on the attack goals, we mainly consider the following two kinds of graph attack methods as our baselines, including *1) Utility attack*: AFGSM [36], TDGIA [47] , and G$^2$A2C [12], and *2) Fairness attack*: FA-GNN [11], FATE [13], and G-FairAttack [41]. The details of these baselines will be further introduced in Appendix F.

**Implementation details.** As shown in Table 1, only a part of the nodes have the label information, and we randomly select 50%, 25%, and 25% labeled nodes as the training set, validation set, and test set, respectively. In line with the prior work, [11], we choose *region* as the sensitive attribute for Pokec-z and Pokec-n, and *gender* for DBLP. For all victim models, we employ a two-layer GCN model as the surrogate model. Due to space limitations, please refer to Appendix G for more reproducibility details.

Table 2: Attack performance of NIFA on different victim GNN models. The results are reported in percentage (%). We **bold** the results when NIFA successfully deteriorates the fairness of victim GNN models (smaller $\Delta_{SP}$ and $\Delta_{EO}$ indicate better fairness, and we aim to maximize the fairness metrics for a fairness attack).

| | | Pokec-z | | | Pokec-n | | | DBLP | | |
|---|---|---|---|---|---|---|---|---|---|---|
| | | Accuracy | $\Delta_{SP}$ | $\Delta_{EO}$ | Accuracy | $\Delta_{SP}$ | $\Delta_{EO}$ | Accuracy | $\Delta_{SP}$ | $\Delta_{EO}$ |
| GCN | before | 71.22±0.28 | 7.13±1.21 | 5.10±1.28 | 70.92±0.66 | 0.88±0.62 | 2.44±1.37 | 95.88±1.61 | 3.84±0.34 | 1.91±0.75 |
| | after | 70.50±0.30 | **17.36±1.16** | **15.59±1.08** | 70.12±0.37 | **10.10±2.10** | **9.85±1.97** | 93.37±1.48 | **13.49±2.83** | **20.33±3.82** |
| GraphSAGE | before | 70.79±0.62 | 4.29±0.84 | 3.46±1.12 | 68.77±0.34 | 1.65±1.31 | 2.34±1.04 | 96.58±0.29 | 4.27±1.09 | 2.78±0.91 |
| | after | 70.05±1.25 | **6.20±1.63** | **4.20±1.77** | 68.93±1.19 | **3.32±1.88** | **3.56±1.91** | 93.92±0.74 | **10.16±2.24** | **16.65±3.30** |
| APPNP | before | 69.79±0.42 | 6.83±1.25 | 5.07±1.26 | 68.73±0.64 | 3.39±0.28 | 3.71±0.28 | 96.58±0.38 | 3.98±1.18 | 2.20±1.08 |
| | after | 69.12±0.70 | **18.44±1.41** | **16.85±1.50** | 67.90±0.76 | **13.47±3.22** | **13.52±3.56** | 92.46±0.94 | **13.88±3.20** | **20.20±4.25** |
| SGC | before | 69.09±0.99 | 7.28±1.50 | 5.45±1.42 | 66.95±1.69 | 2.74±0.85 | 3.21±0.78 | 96.53±0.48 | 4.70±1.26 | 3.11±1.24 |
| | after | 67.83±0.70 | **17.65±1.01** | **16.09±1.06** | 66.72±1.21 | **10.59±2.40** | **10.67±2.61** | 92.56±1.09 | **13.88±3.37** | **20.25±4.44** |
| FairGNN | before | 68.75±1.12 | 1.89±0.63 | 1.51±0.47 | 69.41±0.66 | 1.42±0.35 | 2.32±0.57 | 93.12±1.23 | 1.95±0.99 | 3.09±1.81 |
| | after | 69.38±2.07 | **5.71±2.52** | **4.22±1.89** | 69.97±0.42 | **6.13±5.81** | **6.33±5.77** | 92.56±1.49 | **5.89±2.52** | **10.48±3.82** |
| FairVGNN | before | 68.57±0.45 | 3.79±0.51 | 2.59±0.59 | 67.77±1.00 | 1.90±1.23 | 3.10±1.20 | 95.18±0.54 | 1.90±0.52 | 2.91±1.05 |
| | after | 67.65±0.38 | **11.01±2.79** | **9.28±2.87** | 65.74±1.42 | **3.51±1.51** | **3.65±1.56** | 91.56±1.13 | **7.96±1.49** | **13.57±2.57** |
| FairSIN | before | 67.33±0.22 | 1.73±1.49 | 2.61±1.44 | 67.18±0.30 | 0.39±0.89 | 2.40±1.02 | 94.72±0.62 | 0.23±0.15 | 0.45±0.16 |
| | after | 66.55±0.44 | **9.48±2.62** | **10.39±1.06** | 66.20±0.12 | **11.82±0.75** | **14.58±0.22** | 92.46±0.32 | **10.90±2.12** | **23.65±7.77** |

Table 3: Accuracy and Fairness performance of attack launched by the different attackers. The results are reported in percentage (%). The best attack performance on fairness is **bolded**.

| | Pokec-z | | | Pokec-n | | | DBLP | | |
|---|---|---|---|---|---|---|---|---|---|
| | Accuracy | $\Delta_{SP}$ | $\Delta_{EO}$ | Accuracy | $\Delta_{SP}$ | $\Delta_{EO}$ | Accuracy | $\Delta_{SP}$ | $\Delta_{EO}$ |
| **Clean** | 71.22±0.28 | 7.13±1.21 | 5.10±1.28 | 70.92±0.66 | 0.88±0.62 | 2.44±1.37 | 95.88±1.61 | 3.84±0.34 | 1.91±0.75 |
| *Utility attacks on GNNs* | | | | | | | | | |
| AFGSM | 67.01±0.24 | 3.07±1.67 | 3.45±0.22 | 68.21±0.23 | 5.35±0.15 | 5.68±0.14 | 95.38±0.30 | 5.44±3.48 | 2.78±0.41 |
| TDGIA | 62.20±0.04 | 1.66±0.16 | 0.77±0.10 | 63.57±0.08 | 7.28±0.35 | 6.95±0.34 | 93.42±0.29 | 0.93±0.70 | 1.82±0.87 |
| G$^2$A2C | 39.41±0.94 | 6.89±0.91 | 6.11±0.48 | 34.30±1.71 | 2.23±1.40 | 3.76±1.04 | 86.28±0.25 | 4.21±0.66 | 3.80±0.42 |
| *Fairness attacks on GNNs* | | | | | | | | | |
| FA-GNN | 69.80±0.48 | 6.62±1.21 | 8.67±1.28 | 70.80±0.97 | 2.64±0.76 | 3.45±0.54 | 95.48±0.48 | 3.32±1.65 | 8.74±1.23 |
| FATE | - | - | - | - | - | - | 94.87±0.41 | 3.62±1.49 | 2.12±1.01 |
| G-FairAttack | - | - | - | - | - | - | 95.12±0.38 | 6.80±0.59 | 2.94±1.10 |
| **NIFA** | 70.50±0.30 | **17.36±1.16** | **15.59±1.08** | 70.12±0.37 | **10.10±2.10** | **9.85±1.97** | 93.37±1.48 | **13.49±2.83** | **20.33±3.82** |

## 5.2 Main attack performance

To comprehensively evaluate NIFA's effectiveness, we employ multiple mainstream GNNs including GCN, GraphSAGE, APPNP, and SGC, besides three classical fairness-aware GNNs namely FairGNN, FairVGNN, and FairSIN as our victim models. We record the average accuracy, $\Delta_{SP}$ and $\Delta_{EO}$ before and after conducting our poisoning attack on the victim models five times. The experimental results are reported in Table 2, and we can have the following observations:

- The proposed attack demonstrates consistent effectiveness on all datasets with different mainstream GNNs as victim models. For instance, the $\Delta_{SP}$ and $\Delta_{EO}$ of GCN on Pokec-z increase significantly from 7.13%, 5.10% to 17.36% and 15.59%, respectively. Such observation successfully reveals the vulnerability of GNN fairness under our node injection-based attacks.

- On three fairness-aware models, FairGNN, FairVGNN, and FairSIN, NIFA still causes noticeable fairness impacts. For example, the $\Delta_{EO}$ of FairVGNN on Pokec-z increases from 2.59% to 9.28%, a nearly fourfold increase. It indicates that even fairness-aware GNN models are also vulnerable to our attack, highlighting the urgency of proposing more robust fairness mechanisms.

- Instead of sacrificing the utility of victim GNNs for better fairness attack results, all victim models' accuracy is only slightly impacted on all datasets, which illustrates the distinction between fairness attacks and utility attacks, and underscores NIFA's deceptive nature for administrators.

## 5.3 Comparison with other attack Models

In this section, we aim to compare NIFA with several competitors on graph attacks. Specifically, we choose six well-established attackers on either utility or fairness as our baselines, including AFGSM [36], TDGIA [47], G$^2$A2C [12], FA-GNN [11], FATE [13], and G-FairAttack [41]. For all baselines, the victim model is set as GCN, and the numbers of injected nodes or modified edges are set to be the same as ours for a fair comparison. Note that, both FATE and G-FairAttack fail

to deploy on Pokec-z and Pokec-n due to scalability issues[3]. Results after repeating five times are shown in Table 3.

It can be seen that NIFA consistently achieves the state-of-the-art fairness attack performance on three datasets. The reasons might be two-fold: 1) the utility attack methods are mainly designed to impact the accuracy of victim models, while overlooking the fairness objectives. 2) As pioneering works in fairness attacks on GNNs, all baselines on fairness attacks need to modify the original graph, such as adding or removing some edges or modifying features of real nodes. For deceptiveness consideration, their modifications are usually constrained by a small budget. However, NIFA introduces new nodes into the original graph through node injection and can optimize the injected nodes in a relatively larger feature space. Such superiority of node injection attack helps NIFA have a greater impact on the original graph from the feature perspectives.

## 5.4 Ablation study

In this part, we conduct ablation experiments to prove the effectiveness of the uncertainty-maximization principle, homophily-increase principle, and iterative training strategy, respectively. In detail, we consider the following three variants of NIFA. 1) *NIFA-U*: the uncertainty-maximization principle is removed, and we randomly choose targeted nodes from the labeled nodes. 2) *NIFA-H*: we still choose real nodes with the top k% model uncertainty as the targeted nodes, but the homophily-increase principle is removed, i.e. each injected node may connect with targeted nodes from different sensitive groups simultaneously. 3) *NIFA-I*: we remove the iterative training strategy here, which means that the surrogate model is trained on the clean graph in advance, and the feature optimization process will only involve the training process of injected feature matrix. For all variants, we set GCN as the victim model.

The results are reported in Figure 2, where we can have the following observations. Firstly, after removing the uncertainty-maximization principle (NIFA-U), the fairness attack performance consistently decreases on three datasets. This is expected since the concept of uncertainty helps NIFA find more vulnerable nodes, thus improving the attack

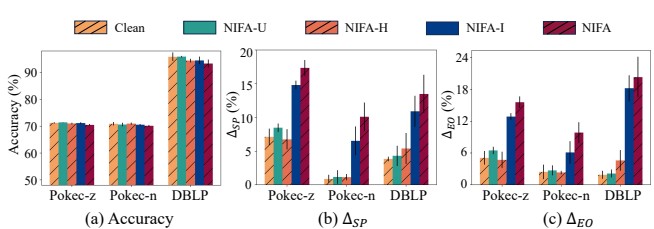

Figure 2: Ablation study of each module in NIFA

effectiveness. Secondly, after removing the homophily-increase principle (NIFA-H), the attack performance drops obviously, which verifies the homophily-ratio is crucial in GNN fairness. Finally, without iterative training during feature optimization (NIFA-I), the attack performance decreases slightly on all datasets. The main reason is that the iterative training strategy could help NIFA to have better robustness to dynamic victim models.

# 6 Defense discussion to fairness attacks on GNNs

As previously emphasized, our intrinsic aim is to unveil the vulnerabilities of existing GNN models in terms of fairness, thereby inspiring related defense research. In fact, as an emerging field that is just beginning to be explored, defense strategies against GNN fairness attacks are relatively scarce. However, we still can summarize several key insights from NIFA for further careful study:

**Reliable training nodes.** One key assumption in NIFA is that the nodes with high model uncertainty will be much easier to be attacked, which can also be supported by the ablation study in Section 5.4. In this way, administrators can pay more attention to these nodes and their abnormal neighbors for defense

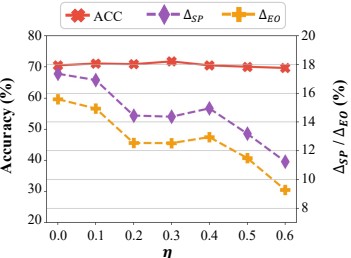

Figure 3: Defense performance on Pokec-z with masking $\eta$ training nodes with the highest uncertainty

---

[3]On Pokec-z and Pokec-n datasets, FATE reports OOM errors and G-FairAttack fails to complete the attack within three days. More scalability analysis will be given in Appendix H.4.

purposes. For example, engineers can pre-train a model to detect the abnormal nodes or edges in advance, especially those that emerged recently in the training data, and weaken their impacts on the model by randomly masking these nodes or edges in the input graph or decreasing their weights in the message propagation during the training of GNNs.

To verify our assumption, we conduct a simple experiment by removing a proportion of nodes ($\eta$) with the highest uncertainty $U$ from supervision signals after the attack. Similarly, GCN [10] is employed as the victim model and we gradually tune $\eta$ from 0 to 0.6 with step 0.1, where $\eta = 0$ means no defense is involved. The performance of NIFA on the Pokec-z dataset with different $\eta$ is illustrated in Figure 3. It can be seen that, since NIFA mainly focuses on attacking nodes with high uncertainty, after masking a part of these nodes during the training stage, the fairness attack performance of NIFA gradually decreases with a small fluctuation in accuracy. However, it is worth noting that although such an intuitive strategy can defend the attack from NIFA to some extent, there is still obvious fairness deterioration compared with the performance of clean GCN in Table 2 ($\Delta_{SP}$=7.13, $\Delta_{EO}$=5.10). More dedicated and effective defense mechanisms in the future are still in demand.

**Strengthen the connections among groups.** One main reason behind the success of NIFA in fairness attack is the guidance of the homophily-increase principle during node injection. The ablation study in Appendix 5.4 also provides empirical evidence for this claim. As we analyze in Section 4.2, NIFA will lead to the increase of node-level homophily-ratio, which means more sensitive-related information will be aggregated and enlarged within the group. Given this, we believe that an effective defensive strategy is to strengthen the information propagation among different sensitive groups, thus preventing the risks of information cocoons [22, 25] and fairness issues.

**Fairness auditing.** At last, we find that a crucial assumption in NIFA and other research [11, 13, 41] is that GNN model administrators will only audit the utility metrics of the models, such as accuracy or F1-scores. Therefore, as long as attackers can ensure that the model utility is not affected excessively, it will be hard for administrators to realize the attack. Consequently, we strongly suggest that model administrators should also incorporate fairness-related metrics into their monitoring scopes, especially before model deployment or during the beta testing phase, thus, mitigating the potential broader negative impacts and social risks. For instance, if an updated GNN model suddenly demonstrates obvious fairness deterioration compared with the previous versions, the model administrators should be careful about the potential fairness attacks. However, the challenge of this approach mainly lies in the diverse definitions of fairness, such as group fairness [5, 23], individual fairness [6], etc., and group fairness based on different sensitive attributes [5, 38] or structures [21, 22] may further lead to different definitions. Therefore, model administrators might need prior knowledge or expertise to determine what kinds of fairness metrics to be included in their monitoring scopes.

## 7 Conclusion

In this work, we aim to examine the vulnerability of GNN fairness under adversarial attacks, thus mitigating the potential risks when applying GNNs in the real world. All existing fairness attacks on GNNs require modifying the connectivity or features of existing nodes, which is typically infeasible in reality. To this end, we propose a node injection-based poisoning attack namely NIFA. In detail, NIFA first proposes two novel principles for node injection operations and then designs multiple objective functions to guide the feature optimization of injected nodes. Extensive experiments on three datasets demonstrate that NIFA can effectively attack most mainstream GNNs and fairness-aware GNNs with an unnoticeable perturbation rate and utility degradation. Our work highlights the vulnerabilities of GNNs to node injection-based fairness attacks and sheds light on future research about robust fair GNNs and defensive mechanisms for potential fairness attacks.

**Limitations.** Firstly, NIFA is still under the settings of gray-box attacks, which requires accessibility to the labels and sensitive attributes. We acknowledge that such information may not always be available and we leave the extensions to the more realistic black-box attack settings as future work. Moreover, although we present some insights on the defense strategies of GNN fairness, more effective defense measures are still under-explored, calling for more future research efforts. At last, currently we only focus on fairness based on sensitive attributes, while neglecting the fairness based on graph structures. Since different fairness may stem from different sources, we leave this as our future work.

## Acknowledgement

The work is supported by the National Natural Science Foundation of China (No.62172174). The authors would also like to thank Xiran Song and Jianxun Lian for their suggestions and contributions.

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

# A    Ethical consideration

In this study, we propose a fairness attack towards GNN models via node injections. It is worth noting that the main purpose of this work is to reveal the vulnerability of current GNN models to fairness attacks, thereby inspiring and motivating both industrial and academic researchers to pay more attention to future potential attacks and enhancing the robustness of GNN fairness. We acknowledge the potential for our research to be misused or exploited by malicious hackers and to have real-world implications or even harm. Therefore, we will open-source our code under the CC-BY-NC-ND license[4] in the future, which means that the associated code cannot be used for any commercial purposes, and no derivatives or adaptations of the work are permitted. Additionally, we discuss some feasible defense mechanisms in Section 6, which we believe can to some extent mitigate the fairness attacks proposed in our work and hopefully inspire future fairness defenses.

# B    Attack settings

In this section, we would like to explicitly introduce our attack settings from the following aspects:

**Attack stage.** Attacks can be categorized into two types according to the time when the attacks take place [42]: *poisoning attack* and *evasion attack*. Poisoning attacks occur at the training phase of victim models, which will lead to poisoned models. In contrast, evasion attacks target the inference phase, and can not affect the model parameters. In this work, NIFA belongs to the poisoning attacks.

**Attacker's knowledge.** Generally, according to the knowledge of attackers, the attack methods can be categorized into three types including white-box attack, black-box attack and gray-box attack [42]. As we introduced in Section 3, we propose NIFA within the gray-box attack settings to make our attack more practical in the real world, which is also consistent with multiple prior research on GNN attacks [11, 32, 41]. Different from white-box attacks and black-box attacks, gray-box attacks mean that attackers can only access the training data, including the input graph $\mathcal{G}$, the labels $\mathcal{Y}$ and the sensitive attribute $s$ of each node. Note that, the model architecture and parameters are invisible to attackers under the gray-box attack settings, which leads to that the attackers need to train a surrogate model in advance to assess the effectiveness of their proposed attacks.

**Attacker's capability.** One merit of NIFA is that there is no need for the attackers to have the authority to modify the existing graph structure, such as adding or deleting edges between existing real nodes, or modifying the existing real nodes' features. In contrast, NIFA injects $\mathcal{V}_I$ malicious nodes and poisons the graph $\mathcal{G}$ into $\mathcal{G}'$ to launch an attack, which is much more practical for the attackers in the real world. For example, in social networks the attackers only need to sign up multiple zombie accounts and interact with real accounts. Note that, different from some injection-based attack [12, 32], NIFA will not modify the training set and true node label set $\mathcal{Y}$, as such operations are typically infeasible in the real world. The intrinsic idea of NIFA is to impact the GNN training process through massage propagation on a poisoned graph.

Within the gray-box attack settings, we also assume that the attackers have sufficient computational resources and budget to train a surrogate model and have access to the real graph as input. Similar to prior attack methods [32], attackers are also required to set thresholds $b$ and $d$ for the number of injected nodes and their degrees respectively to make NIFA deceptive and unnoticeable.

# C    Proof

**Lemma 1.** *For target node $u$ that will connect with injected nodes, our proposed node injection strategy will lead to the increase of node-level homophily-ratio $\mathcal{H}_u$.*

*Proof.* Given a target node $u$ with $k$ neighbors that have the same sensitive attribute with $u$, we simply assume it will connect with $n$ injected nodes after node injection. Since all injected nodes in our proposed node injection belong to the same sensitive attribute as target node $u$, then the node-level homophily-ratio after node injection $\mathcal{H}'_u$ is:

$$\mathcal{H}'_u = \frac{k+n}{|\mathcal{N}_u|+n} \geq \frac{k}{|\mathcal{N}_u|} = \mathcal{H}_u \tag{16}$$

---

---
**Algorithm 1** Training process of NIFA
---
**Input:** Graph $\mathcal{G}$. Hyper-parameters: node budget $b$, edge budget $d$, $\alpha$ and $\beta$, learning rates $\gamma^S$, $\gamma^F$.
**Output:** Poisoned graph $\mathcal{G}'$.
 1: Initialize Bayesian GNN's parameter $\theta_\mathcal{B}$, surrogate model's parameter $\theta_\mathcal{S}$ and injected nodes'
    feature matrix $\mathbf{X}_I$.
 2: Train the Bayesian GNN by Eq. (8) and Eq. (9).
 3: Estimate the node uncertainty $U$ in the original graph $\mathcal{G}$ by Eq. (10) and select the target nodes.
 4: Connect the injected nodes with target nodes according to the homophily-increase principle.
 5: **for** $iter = 1$ **to** $max\_iter$ **do**
 6:   **for** $step = 1$ **to** $max\_step$ **do**
 7:     Compute cross-entropy loss $L_{CE}$ for $\mathcal{S}$ by Eq. (11).
 8:     Update surrogate model: $\theta_\mathcal{S} \leftarrow \theta_\mathcal{S} + \gamma^S \cdot \nabla_{\theta_\mathcal{S}} L_{CE}$.
 9:   **end for**
10:   **for** $step = 1$ **to** $max\_step$ **do**
11:     Compute $L$ by Eq. (15).
12:     Update injected feature: $\mathbf{X}_I \leftarrow \mathbf{X}_I + \gamma^F \cdot \nabla_{\mathbf{X}_I} L$.
13:   **end for**
14:   Clamp $\mathbf{X}_I$ between min and max of $\mathbf{X}$.
15: **end for**
16: **if** $\mathbf{X}$ is discrete **then**
17:   Round $\mathbf{X}_I$ into integer.
18: **end if**
19: **return** $\mathcal{G}'$
---

Such inequality holds true when $|\mathcal{N}_u| \geq k$. $\qquad\qquad\qquad\qquad\qquad\qquad\qquad\qquad\qquad\quad\square$

## D   Implementation algorithm

Due to space limitation, here we provide the complete training process of NIFA in Algorithm 1. The training process and evaluation process are also literally described in Section 4.4.

## E   Additional descriptions on victim models

In this part, we give an introduction to the victim models used in our experiment.

- **GCN** [10]: Borrowing the concept of convolution from the computer vision domain, GCN employs convolution operation on the graph from a spectral perspective to learn the node embeddings.
- **GraphSAGE** [14]: Given the potential neighborhood explosion issues in GCN, GraphSAGE samples a fixed number of neighbors at each layer during neighborhood aggregation, which greatly improves the training efficiency.
- **APPNP** [15]: Inspired by PageRank, APPNP decouples the prediction and propagation in the training process, which resolves inherent limited-range issues in message-passing models without introducing any additional parameters.
- **SGC** [39]: SGC empirically finds the redundancy of non-linear activation function, and achieves comparable performance with much higher efficiency.
- **FairGNN** [5]: Through proposing a sensitive attribute estimator and an adversarial learning module, FairGNN maintains high classification accuracy while reducing unfairness in scenarios with limited sensitive attribute information.
- **FairVGNN** [38]: FairVGNN discovers the leakage of sensitive information during information propagation of GNN models, and generates fair node features by automatically identifying and masking sensitive-correlated features.
- **FairSIN** [40]: Instead of filtering out sensitive-related information for fairness, FairSIN deploys a novel sensitive information neutralization mechanism. Specifically, FairSIN will learn to introduce additional fairness facilitating features (F3) during message propagation to neutralize sensitive information while providing more non-sensitive information.

# F  Additional descriptions on baselines

In this section, we will introduce more details about the baselines used in our experiments.

- **AFGSM** [36]: As a node injection-based attack, AFGSM designs an approximation strategy to linearize the victim model and then generates adversarial perturbation efficiently. In general, AFGSM is scalable to much larger graphs.

- **TDGIA** [47]: Aiming at attacking the model performance, TDGIA first introduces a topological edge selection strategy to select targeted nodes for node injection, and then generate injected features through smooth feature optimization.

- **G$^2$A2C** [12]: Similar to NIPA [32], G$^2$A2C also proposes a node injection attack through reinforcement learning – Actor Critic. Specifically, G$^2$A2C devises three core modules including *Node Generator*, *Edge Sampler*, *Value Predictor* to model the full process of node injection.

- **FA-GNN** [11]: To the best of our knowledge, FA-GNN is the first work to conduct a fairness attack on GNNs. In detail, FA-GNN empirically discovers several edge injection strategies, which could impact the GNN fairness with slight utility compromise.

- **FATE** [13]: FATE is a meta-learning-based fairness attack framework for GNNs. To be concrete, FATE formulates the fairness attacks as a bi-level optimization problem, where the lower-level optimization guarantees the deceptiveness of an attack while the upper-level optimization is designed to maximize the bias functions.

- **G-FairAttack** [41]: As a poisoning fairness attack, G-FairAttack consists of two modules, including a surrogate loss function and following constrained optimization for deceptiveness. Like FATE and FA-GNN, G-FairAttack also belongs to graph modification attacks, i.e., the original link structure between existing nodes will be modified during the attack.

# G  Reproducibility details

The implementation details are first provided in Section 5.1 in the original paper. Here we would like to provide more implementation details from the following four aspects:

**Environment**. All experiments are conducted on a server with Intel(R) Xeon(R) Gold 5117 CPU @ 2.00GHz and 32 GB Tesla V100 GPU. The experimental environment is based on Ubuntu 18.04 with CUDA 11.0, and our implementation is based on Python 3.8 with PyTorch 1.12.1 and Deep Graph Library (DGL) 1.1.0.

**Victim models.** For all victim models, we set the learning rate as 0.001, and the hidden dimension as 128 after careful tuning. For most victim models, the dropout ratio is set to 0 by default. Specifically, for GCN and GraphSAGE, the layer number is set to 2, and we employ mean pooling aggregation for GraphSAGE. For SGC, the hop-number $k$ is set to 1. For APPNP, following the suggestions in the official paper, the teleport probability $\alpha$ is set as 0.2, and we set the iteration number $k$ as 1 after careful tuning. For FairGNN, we employ the GAT as the backbone model, which shows a better performance in the original paper, and set the dropout ratio as 0.5. The objective weights $\alpha$ and $\beta$ are set as 4 and 0.01, respectively. For FairVGNN, GCN is set to be the backbone model, and we set the dropout ratio as 0.5. The prefix cutting threshold $\epsilon$ is searched from $\{0.01, 0.1, 1\}$, and the mask density $\alpha$ is 0.5. The epochs for the generator, discriminator, and classifier are selected from $\{5, 10\}$ as suggested in the official implementation. For FairSIN, we also utilize GCN as the backbone model, and we set the weight of neutralized feature $\delta$ as 4 after tuning, and set the hidden dimension as 128 for a fair comparison. All learning rates involved in FairSIN are set to 0.001 after careful fine-tuning and other hyper-parameters are set according to the official implementations[5].

**Baselines details.** For all baselines that involve graph structure manipulation, the numbers of injected nodes and edges on three datasets are made identical to the settings of our model for a fair comparison. For methods that only inject new nodes, such as AFGSM, TDGIA and G2A2C, we require the number of injected nodes and average degree of injected nodes to be the same as ours. For methods that require to modify the graph structure, such as FA-GNN, FATE and G-FairAttack, we set the number of modified edges to be the same with our injected edges, i.e. the added edges between injected nodes and original nodes. To be concrete, for the AFGSM, across all datasets,

---

[5]https://github.com/BUPT-GAMMA/FairSIN/

we utilize the direct attack setting and allow the model to perturb node features. For TDGIA, the weights $k_1$, and $k_2$, which are used for calculating topological vulnerability are set to 0.9 and 0.1, respectively. For G$^2$A2C, the temperature of Gumbel-Softmax is set to 1.0, the discount factor is set to 0.95, and the Adam optimizer is utilized with a learning rate of $10^{-4}$. Furthermore, we adopt early stopping with a patience of three epochs. For FA-GNN, we utilize the $DD$ strategy, which has the best attack performance in the original paper. For FATE, the perturbation mode is *filp* and the attack step is set to 3 as the official implementation[6] suggested. For G-FairAttack, the proportion of candidate edges is set to 0.0001 for fast computation on our datasets, and we follow the default settings in the official repository[7] for other hyper-parameters. It is worth noting that, G-FairAttack can be utilized as either an evasion attack or a poisoning attack according to the original paper, and we follow the poisoning attack settings to be the same with NIFA.

**Details for implementing NIFA.** We employ a two-layer GCN model as the surrogate model, whose hidden dimension is set to 128, and the dropout ratio is set to 0. The learning rates for optimizing the surrogate model and injected features are both set to 0.001 after tuning. The sampling times $T$ of the Bayesian Network is 20. The objective weights $\alpha$ and $\beta$ for all datasets are searched from $\{0.005, 0.01, 0.02, 0.05, 0.1, 0.2\}$ and $\{2, 4, 8, 16\}$, respectively. The number of injected nodes is set to 1% of the number of labeled nodes in the original graph, and the degree of injected nodes $d$ is set to be 50, 50, and 24 on three datasets, respectively, which are the

Table A1: Hyper-parameters statistics

| Notations | Pokec-z | Pokec-n | DBLP |
|---|---|---|---|
| $\alpha$ | 0.01 | 0.01 | 0.1 |
| $\beta$ | 4 | 4 | 8 |
| $b$ | 102 | 87 | 32 |
| $d$ | 50 | 50 | 24 |
| $k$ | 0.5 | 0.5 | 0.5 |
| $max\_step$ | 50 | 50 | 50 |
| $max\_iter$ | 20 | 20 | 10 |

average node degrees in the original graph. As for the uncertainty threshold $k\%$, we search in a range of $\{0.1, 0.25, 0.5, 0.75\}$. The hyper-parameter analysis will be further elaborated in Appendix H.3. The $max\_iter$ and $max\_step$ in Algorithm 1 and other proposed hyper-parameters for each dataset are summarized in Table A1.

Table A2: Comparison of the statistics on the clean graph (Perturbation rate is 0.00) and the graphs poisoned by NIFA with different perturbation rates. For better illustration, we also provide the relative rate of change ($|\Delta|$) in the table.

| | Perturbation Rate | Gini Coefficient | | Assortativity | | Power Law Exp. | | Triangle Count | | Rel. Edge Distr. Entropy | | Characteristic Path Length | |
|---|---|---|---|---|---|---|---|---|---|---|---|---|---|
| | | value | $|\Delta|$ | value | $|\Delta|$ | value | $|\Delta|$ | value | $|\Delta|$ | value | $|\Delta|$ | value | $|\Delta|$ |
| **Pokec-z** | 0.00 | 0.5719 | 0 | 0.2108 | 0 | 1.6152 | 0 | 767,688 | 0 | 0.0259 | 0 | 4.5208 | 0 |
| | 0.01 | 0.5696 | 0.41% | 0.2100 | 0.40% | 1.6103 | 0.30% | 767,787 | 0.01% | 0.0259 | 0.13% | 4.5062 | 0.32% |
| | 0.02 | 0.5677 | 0.75% | 0.2093 | 0.74% | 1.6064 | 0.54% | 767,917 | 0.03% | 0.0259 | 0.24% | 4.4951 | 0.57% |
| | 0.05 | 0.5627 | 1.61% | 0.2071 | 1.79% | 1.5967 | 1.15% | 768,287 | 0.08% | 0.0260 | 0.42% | 4.4732 | 1.05% |
| **Pokec-n** | 0.00 | 0.5634 | 0 | 0.1978 | 0 | 1.6609 | 0 | 531,590 | 0 | 0.0296 | 0 | 4.6365 | 0 |
| | 0.01 | 0.5613 | 0.37% | 0.1958 | 0.96% | 1.6557 | 0.31% | 531,695 | 0.02% | 0.0296 | 0.08% | 4.6215 | 0.32% |
| | 0.02 | 0.5595 | 0.68% | 0.1939 | 1.96% | 1.6513 | 0.58% | 531,776 | 0.03% | 0.0296 | 0.13% | 4.6103 | 0.56% |
| | 0.05 | 0.5555 | 1.39% | 0.1895 | 4.15% | 1.6409 | 1.20% | 532,089 | 0.09% | 0.0297 | 0.20% | 4.5902 | 1.00% |
| **DBLP** | 0.00 | 0.4137 | 0 | 0.1335 | 0 | 2.0730 | 0 | 73,739 | 0 | 0.0820 | 0 | 6.4867 | 0 |
| | 0.01 | 0.4137 | 0.01% | 0.1316 | 1.44% | 2.0618 | 0.54% | 73,744 | 0.01% | 0.0818 | 0.27% | 6.3627 | 1.87% |
| | 0.02 | 0.4148 | 0.27% | 0.1304 | 2.32% | 2.0528 | 0.98% | 73,751 | 0.02% | 0.0815 | 0.69% | 6.2886 | 3.02% |
| | 0.05 | 0.4208 | 1.71% | 0.1311 | 1.83% | 2.0307 | 2.04% | 73,774 | 0.05% | 0.0801 | 2.34% | 6.1789 | 4.72% |

# H   Additional experiments

## H.1   In-depth analysis of the poisoned graph

As a poisoning attack, it is crucial to ensure that after the attack, the poisoned graph's characteristics should remain similar to that of the clean graph. Otherwise, administrators can easily notice the attack through abnormal graph structures or node features. To this end, we conduct an in-depth analysis of the poisoned graph by NIFA from the following two perspectives:

---

[6]https://github.com/jiank2/FATE

[7]https://github.com/zhangbinchi/G-FairAttack

**Structural analysis.** Similar to prior work [2, 32], we investigate several key graph characteristics in this section, including *Gini Coefficient*, *Assortativity*, *Power Law Exponent*, *Triangle Count*, *Relative Edge Distribution Entropy* and *Characteristic Path Length*, whose definitions and implementations can be found here[8]. The graph statistics under different perturbation rates are shown in Table A2. It can be concluded that: (1) Thanks to the low perturbation rate required by NIFA, the poisoned graphs share quite similar characteristics with clean graphs. Under most cases, the relative rate of change $|\Delta|$ is smaller than 1%, especially when the perturbation rate is small. (2) With the increase of perturbation rate, the poisoning attack becomes more obvious, which can be verified by the increased $|\Delta|$ on all graph structure statistics. This observation further supports the necessity of requesting a low perturbation rate for an attack. (3) Several key statistics showcase a consistent trend across the three datasets as the perturbation rate increases. For example, with the increase in attack intensity, the degree assortativity consistently decreases on three datasets, indicating that there are more connections between nodes with significantly different degrees due to the injection of malicious nodes. Similar observations can also be found in key statistics like *Triangle Count*, *Gini Coefficient* and *Characteristic Path Length*, which implies that these statistics can be potentially utilized for fairness attack defense and auditing.

**Feature analysis.** Besides graph structural analysis, we also conduct experiments to analyze the nodes' features of the poisoned graph. In detail, after conducting fairness attacks through NIFA, for both labeled nodes in the real graph and injected nodes, we illustrate their features' visualization results based on t-SNE [35] in Figure A1. Note that, since the number of injected nodes is much smaller than that of the original labeled nodes, we slightly increase the scatter size of injected nodes for better visualization.

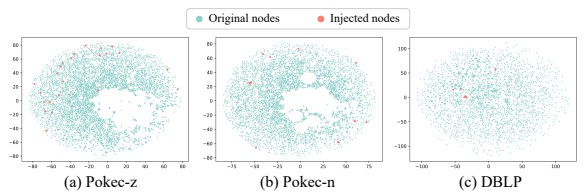

Figure A1: T-SNE visualization of poisoned graph's node features

It can be seen that, 1) the feature layouts of Pokec-z and Pokec-n are quite similar, since both datasets are sampled from the same social graph. 2) On all three datasets, the distributions of injected nodes by NIFA are relatively diverse, which have no obvious patterns, and are hard to recognize from the original labeled nodes. Such observation further verifies the feature deceptiveness of NIFA.

### H.2 Alternative analysis in node selection

During the node injection process, NIFA proposes an uncertainty-maximization principle and selects target nodes with the highest model uncertainty score. In fact, besides estimating model uncertainty, there are other alternative methods for finding vulnerable nodes. For example, TDGIA [47] introduces the concept of "topological vulnerability",

Table A3: Attack performance (%) with different target node selection strategies. The best attack performance is **bolded**.

|  | **Pokec-z** | | | **Pokec-n** | | | **DBLP** | | |
|---|---|---|---|---|---|---|---|---|---|
|  | Acc. | $\Delta_{SP}$ | $\Delta_{EO}$ | Acc. | $\Delta_{SP}$ | $\Delta_{EO}$ | Acc. | $\Delta_{SP}$ | $\Delta_{EO}$ |
| **Clean** | 71.22 | 7.13 | 5.10 | 70.92 | 0.88 | 2.44 | 95.88 | 3.84 | 1.91 |
| **Degree** | 70.50 | 15.76 | 14.01 | 69.77 | **12.39** | **11.93** | 94.72 | 6.30 | 15.10 |
| **Uncertainty** | 70.50 | **17.36** | **15.59** | 70.12 | 10.10 | 9.85 | 93.37 | **13.49** | **20.33** |

and selects nodes with low degrees as target nodes. In this part, we also conduct experiments with selecting target nodes with the lowest degree as a variant of NIFA. The victim model is GCN.

The results are shown in Table A3. It can be seen that, degree-based node selection (denoted as "Degree") also achieves promising attack performance compared with NIFA (denoted as "Uncertainty"), which indicates that model uncertainty is not the only feasible criterion for target node selection. However, the degree-based selection may be ineffective when the graph is extremely dense and has few low-degree nodes, which deserves more careful study in the future.

### H.3 Hyper-parameter analysis

To better understand the different roles of hyper-parameters in NIFA, we study the impact of $\alpha$, $\beta$, $b$, and $k$ in this part where GCN is employed as the victim model.

---

[8]https://github.com/danielzuegner/netgan/tree/master

**The impact of $\alpha$ and $\beta$.** As the weights of objective functions in Eq. (15), the impacts of $\alpha$ and $\beta$ are illustrated in Figure A2 and Figure A3, respectively. For three datasets, they perform best at different $\alpha$ and $\beta$ values, indicating that the appropriate values of $\alpha$ and $\beta$ depend on the datasets. In contrast, the accuracy remains relatively stable with changes in $\alpha$ and $\beta$, which indicates that our attack is utility-friendly.

**The impact of node injection budget $b$.** We illustrate the impact of the node injection budget $b$ in Figure A4, where the x-axis denotes the perturbation rate, i.e. the proportion of $b$ to the number of labeled nodes in the original graph. As expected, the increase of $b$ leads to better fairness attack performance in most cases with more obvious accuracy compromise. Empirically, 0.01 will be a near-optimal choice while being unnoticeable.

**The impact of uncertainty threshold $k$.** The impact of $k$ is illustrated in Figure A5, where we tune $k\%$ in a set of $\{0.1, 0.25, 0.5, 0.75\}$. It can be seen

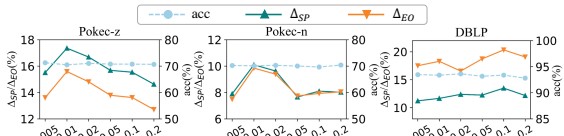

Figure A2: The impact of $\alpha$ on three datasets

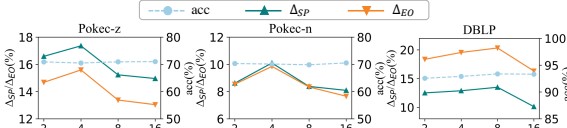

Figure A3: The impact of $\beta$ on three datasets

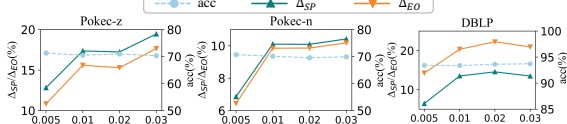

Figure A4: The impact of perturbation rate on three datasets

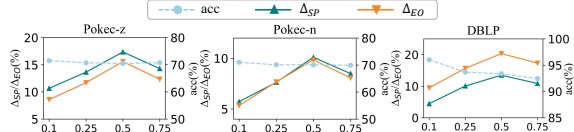

Figure A5: The impact of k% on three datasets

that all datasets show a similar preference for $k\%$ with an optimal value of 0.5. Intuitively, with a higher $k\%$, it is hard for NIFA to attack the target nodes with low uncertainty, whereas, with a lower $k\%$, the impact of the attack will be limited by the insufficient number of targeted nodes.

## H.4 Efficiency analysis

To evaluate the efficiency and scalability of NIFA, we would like to compare the attack time cost and memory cost between NIFA and two competitive baselines including FATE [13] and G-FairAttack [41]. To be fair, the attack budgets for the three models are set to be the same in advance. We report their memory cost and time cost for finishing

Table A4: Efficiency Analysis of NIFA

|  | **Pokec-z** | | **Pokec-n** | | **DBLP** | |
|---|---|---|---|---|---|---|
|  | Time | Memory | Time | Memory | Time | Memory |
| **FATE** | – | OOM | – | OOM | 87.13 s | 32258 MB |
| **G-FairAttack** | >72 h | 7865 MB | >72 h | 7329 MB | 93048.20 s | 2445 MB |
| **NIFA** | 137.04 s | 4319 MB | 167.07 s | 4213 MB | 127.52 s | 5829 MB |

poisoning attacks on Pokec-z, Pokec-n and DBLP in Table A4. The environment configurations for our experiments are introduced in Section G. It can be seen that, NIFA can be successfully deployed on three datasets with acceptable time cost and memory cost. In contrast, both FATE and G-FairAttack face scalability issues, especially when facing large graphs such as Pokec-z and Pokec-n. As shown in Table 1, both Pokec-z and Pokec-n have much more edges compared with DBLP, which causes FATE to report OOM errors and G-FairAttack to fail to finish the attack within 72 hours.

## H.5 Robustness to defense strategies

Besides the defense discussions in Section 6, we analyze the effectiveness of conventional defense strategies in this section. In detail, two classic GNN defense models – GNNGuard [44] and ElasticGNN [18] are deployed to test their defense capabilities against NIFA. It is worth noting that, both defense models are originally designed for the attacks on prediction utility. We conduct experiments on three datasets, and the victim model is GCN as default. As shown in Table A5, both defense models only maintain the utility performance and failed to fully eliminate the impact of fairness at-

Table A5: The performance of defense strategies against NIFA.

| | Pokec-z | | | Pokec-n | | | DBLP | | |
|---|---|---|---|---|---|---|---|---|---|
| | **Acc** | $\Delta_{SP}$ | $\Delta_{EO}$ | **Acc** | $\Delta_{SP}$ | $\Delta_{EO}$ | **Acc** | $\Delta_{SP}$ | $\Delta_{EO}$ |
| **Clean** | 71.22±0.28 | 7.13±1.21 | 5.10±1.28 | 70.92±0.66 | 0.88±0.62 | 2.44±1.37 | 95.88±1.61 | 3.84±0.34 | 1.91±0.75 |
| **NIFA** | 70.50±0.30 | 17.36±1.16 | 15.59±1.08 | 70.12±0.37 | 10.10±2.80 | 9.85±1.97 | 93.37±0.28 | 13.49±2.83 | 20.33±3.82 |
| **GNNGuard** | 71.50±0.35 | 18.13±2.31 | 16.32±1.44 | 70.55±0.43 | 13.82±1.76 | 13.38±1.42 | 95.73±0.28 | 7.42±1.82 | 14.94±1.58 |
| **ElasticGNN** | 71.36±0.20 | 13.53±0.92 | 11.36±1.23 | 70.32±0.28 | 6.90±1.02 | 6.44±1.32 | 94.22±0.37 | 9.76±1.65 | 17.04±2.01 |

tacks. For example, compared with the performance after the attack, although ElasticGNN reduces the $\Delta_{SP}$ from 17.36% to 13.53%, the fairness is still worse than that before the attack – 7.13%. We believe that the main reason behind such observation is that all these defense methods are designed for utility-targeted attacks, whose objectives are totally different from fairness attacks like NIFA. Such observations also indicate that more effective defense mechanisms are still in demand for fairness attacks on GNNs.

## H.6  Performance with limited training ratio

As introduced in Section 5.1, our datasets are all collected from the previous work – FA-GNN [11], and our train / val / test ratios are set to be consistent with its settings, which is 50% / 25% / 25%, respectively. However, in more realistic scenarios the percentage of labeled training nodes might be significantly smaller due to the heavy cost of annotation. To evaluate the effectiveness of NIFA under such settings, we

Table A6: The performance of NIFA with more limited training nodes. The victim model is GCN.

| | | Pokec-z | | | Pokec-n | | |
|---|---|---|---|---|---|---|---|
| | | **Acc** | $\Delta_{SP}$ | $\Delta_{EO}$ | **Acc** | $\Delta_{SP}$ | $\Delta_{EO}$ |
| 25% | before | 71.47±0.59 | 6.26±1.63 | 4.23±1.55 | 70.16±0.45 | 1.03±0.70 | 2.87±0.53 |
| | after | 70.48±0.41 | **15.16±1.64** | **13.18±1.77** | 69.49±0.31 | **9.79±2.02** | **9.52±1.84** |
| 10% | before | 70.40±0.28 | 6.29±2.77 | 4.63±2.43 | 70.25±0.57 | 2.56±1.32 | 3.72±0.32 |
| | after | 70.14±0.26 | **15.26±3.02** | **13.36±3.06** | 69.25±0.62 | **11.94±1.78** | **12.00±1.75** |
| 5% | before | 70.03±0.54 | 4.90±1.98 | 3.75±1.01 | 68.61±1.15 | 3.79±2.30 | 4.21±1.94 |
| | after | 69.49±0.65 | **14.35±2.03** | **12.42±2.04** | 68.05±1.63 | **10.31±5.68** | **10.60±5.04** |

decrease the training ratio from 50% to 25%, 10% and 5%, respectively. The attack performance of NIFA on Pokec-z and Pokec-n is shown in Table A6. It can be seen that, even with much fewer labeled training nodes, NIFA still consistently demonstrates promising attack performance.

