# OpenReview forum: "Are Your Models Still Fair? Fairness Attacks on Graph Neural Networks via Node Injections"
_NeurIPS.cc/2024/Conference — NeurIPS 2024 poster_

### Official Review · Reviewer_wXHX · 2024-06-19

**Soundness:** 3
**Presentation:** 3
**Contribution:** 2
**Rating:** 5
**Confidence:** 5

**Summary:**

The authors propose a fairness attack on GNN through node injection. They propose two node injection principles, the uncertainty-maximization and homophily-increase principle, to make fake node injections lead to a more significant fairness compromise.

**Strengths:**

This article is well-written and highly readable.

The focus on attacking fairness is interesting.

The author’s discussion on potential extensions of the method, such as different approaches to measure node vulnerability and potential mitigation methods, is encouraging.

The choice of datasets and baselines for validating attack performance is representative, demonstrating the significant effectiveness of the proposed method.

**Weaknesses:**

1. The main concern is that the proposed node-injection-based fairness attack could potentially be mitigated by existing defenses designed for accuracy-based node-injection attacks. The distinction between fairness-targeted and accuracy-targeted attacks is not discussed, and there is a lack of an in-depth discussion on related work concerning node-injection attacks aimed at accuracy.

2. Another concern is that the theoretical effectiveness of "the node injection strategy is evaluated by an increase in the node-level homophily ratio". This homophily ratio does not clearly establish a connection with fairness metrics, i.e., DP/EO.  Providing a theoretical guarantee that the proposed node injection strategy results in more significant improvements in DP/EO compared to random node injection will enhance the validity.

3. Additionally, the motivation of attackers to undermine fairness is not clearly discussed. Adding some real-world examples to demonstrate the motivation of such attacks will be better.

**Questions:**

In addition to the weaknesses listed above, my questions for the authors include:

1. The injected node features require the attacker to perform local training on a given training node set. Can the attacker  obtain the training node set, especially when it contains private information?

2. In the discussion of mitigation methods, such as Reliable Training Nodes and Strengthening Connections Among Groups, the authors focus on general mitigation strategies. These strategies are also suitable for defending against accuracy-targeted node-injection attacks and lack a discussion on the specifics of fairness-targeted attacks. This raises the question of whether existing defenses against accuracy-targeted attacks are sufficient to defend against fairness-targeted attacks.

3. The experimental results in Table 2 show that fairness is compromised while accuracy also decreases, which contradicts the expected trade-off between fairness and accuracy. The authors should explain the reason for this occurrence. In principle, when fairness is compromised with the goal of minimizing accuracy loss, accuracy should improve.

As the topic is interesting, I am willing to adjust my score based on the authors' responses.

**Limitations:**

1. The proposed method is designed for binary classification with two sensitive attributes. It should be considered whether and how the proposed method can be extended to multi-class classification and multiple sensitive attributes.

2. The experiments were conducted on a two-layer GCN, which is consistent with the baseline [10]. However, a discussion is expected about the feasibility of the attack  on GCNs with different architectures.

---

> ### Author Rebuttal · Authors · 2024-08-05
>
> **[Concerns about the discussion about node-injection-based attack (W1)]**
>
> The main distinction between fairness-targeted and accuracy-targeted attacks is the different attack objectives. The accuracy-targeted attacks aim to undermine the model accuracy, while fairness-targeted attacks aim to deteriorate the model fairness without significantly compromising the model accuracy. The results in Table 3 in our manuscript also support this claim. Although we introduce several node-injection-based attacks in Section 2, we agree that an in-depth review of corresponding related work would be better, and we will add more discussion on node-injection-based attacks to our manuscript in the future.
>
> ------
>
> **[Concerns about the defense strategies (W1, Q2)]**
>
> Thanks for your insightful comments! We hope to address your concerns on the defense strategies with the following responses:
>
> 1. While selecting more reliable training nodes might help resist accuracy-targeted adversarial attacks, our experiments in Figure 2 actually show that this approach **does not completely mitigate** the fairness attacks posed by NIFA. More effective methods are still in demand.
> 2. We further test the defense capabilities of two classic GNN defense models—GNNGuard [4] and ElasticGNN [5]—against NIFA. The results are shown **in Table R5 in the uploaded PDF**. As shown in the results, **both defense models only maintain the utility performance and failed to fully eliminate the impact of fairness attacks.**
>
> ------
>
> **[Concerns about the Lemma 1 (W2)]**
>
> As we claimed in footnote 2, the relationship between homophily ratio and fairness is widely discussed in previous work. To better understand the role of homophily ratio in GNN fairness and mitigate the gap, we would like to provide more theoretical analysis. **Due to space limitation, we place the complete theoretical analysis in the global rebuttal box**.
>
> ------
>
> **[Concerns about the motivation (W3)]**
>
> In fact, fairness attacks on GNNs have numerous potential applications. Besides the example of GNN-based recommendation in the introduction, professional social networks like LinkedIn also face some potential risks. Specifically, when GNNs are used to identify high-potential candidates, attackers might use fake accounts to manipulate the model into predicting high potential for their group with a much higher probability, securing better job offers while harming other demographic groups.
>
> ------
>
> **[Concerns about the capability of attackers (Q1)]**
>
> As we introduced in Appendix B, NIFA is under the gray-box attack settings, which is realistic and widely studied in previous utility attacks of GNNs [1-3]. In gray-box attack settings, the attackers have access to the training data, including graph, node attributes and training labels, but cannot see the model architecture and parameters. In fact, the training data including node attributes and ground truth labels are not hard to get in the real world. For example, some user attributes like gender, region are actually public on some social platforms like Weibo, LinkedIn or Twitter.
>
> ------
>
> **[Concerns about the the accuracy (Q3)]**
>
> Although many studies in pursuing GNN fairness suggest a trade-off between utility and fairness—where better fairness often results in lower utility—we believe this trade-off does not fully hold in the context of fairness attacks. For instance, if a GNN model's predictions are entirely aligned with sensitive attributes, resulting in a 100% SP (Statistical Parity), its accuracy would inevitably suffer.
>
> Additionally, we would like to further discuss some **potential methods to better alleviate the utility decrease introduced by NIFA**, such as controlling the number of target nodes, i.e. the nodes with the top $k$ highest uncertainty. Intuitively, with more nodes connected with injected nodes, the utility will be more likely to be influenced. To support our claims, we tune the $k$ in a range of {0.10, 0.25, 0.50, 0.75} in DBLP, and the attack performance on GCN and GraphSAGE is shown **in Table R1 in the uploaded PDF**. It can be seen that, after decreasing $k$, the utility after the attack can be better preserved.
>
> ------
>
> **[Concerns about the expandability (L1)]**
>
> Thanks for your inspiring question, and we would like to discuss the expandability of NIFA from the following two perspectives:
>
> - **Multi-class classification**: In fact, our method can naturally fit the multi-label classification scenarios, since there is no specific requirement on the number of categories in our framework, which can also be verified by the problem definitions in Section 3.  Since our datasets are collected from the prior work -- FA-GNN, we mainly focus on binary classification tasks to be consistent.
> - **Multiple sensitive attributes**: As we claimed in Section 3, NIFA can be easily extended to multiple sensitive attributes. The only distinction is that the attackers need to equally partition the injected nodes into multiple groups instead of two groups during the node injection process, and make sure each injected node will only connect to real nodes with the same sensitive attribute.
>
> ------
>
> **[Concerns about the GCN architectures (L2)]**
>
> Thanks for your suggestions. We further conduct model architecture analysis for GCN by modifying the model layer and hidden dimension, and the attack results on Pokec-z and Pokec-n are shown **in Table R6 in the uploaded PDF**. It can be seen that, NIFA demonstrates promising robustness towards the GCN models with different model layers and hidden dimensions.
>
>
> **References**
>
> [1] Adversarial Attacks on Graph Neural Networks via Meta Learning, ICLR 2019.
>
> [2] A Unified Framework for Data Poisoning Attack to Graph-based Semi-supervised Learning, NeurIPS 2019.
>
> [3] Adversarial Attacks on Fairness of Graph Neural Networks, ICLR 2024.
>
> [4] GNNGUARD: Defending Graph Neural Networks against Adversarial Attacks, NeurIPS 2020.
>
> [5] Elastic Graph Neural Networks, ICML 2021.

---

> > ### Comment · Reviewer_wXHX · 2024-08-09
> >
> > I would like to thank the authors for their response and clarifications. At this point, I have no further questions. Although the current work still relies on a gray-box attack model and lacks an effective defense mechanism against fairness attacks, I acknowledge these are existing challenges and appreciate the authors' clarification of these limitations. I increased my score based on new experiments and explanations of convincing motivation, better robustness compared to accuracy attacks, theoretically supported proxy metric, and generalizability.

---

> > > ### Author Response · Authors · 2024-08-09
> > > **Thank you for the comments**
> > >
> > > Thank you again for your valuable review and reconsideration of scores!

---

### Official Review · Reviewer_U6JX · 2024-07-04

**Soundness:** 2
**Presentation:** 3
**Contribution:** 1
**Rating:** 4
**Confidence:** 5

**Summary:**

The authors propose a Fairness GNN Attack method called Node Injection-based Fairness Attack (NIFA). The proposed method aims to increase the bias of GNN models by injecting nodes into the graph. NIFA identifies nodes with high uncertainty to target them, then connects the injected nodes in such a way increasing the homophily of the graph. The features of the injected nodes are then optimized to balance utility with fairness. NIFA is evaluated on multiple benchmark datasets and compared with Fairness GNN attack methods.

**Strengths:**

- The paper is very well written and easy to follow.
- The proposed method seems to be technically sound.
- The fairness of GNN models and Fairness attack methods are both important and well motivated problems.

**Weaknesses:**

- The proposed attack method is not well motivated, could the authors provide concrete application domains/cases where it is not possible to modify/attack the existing edges between nodes in the graph but it is possible to inject nodes into it and connect such nodes to the rest of the graph?

- Furthermore, if modifying the edges in graph is considered unrealistic, how is this different from the part in the proposed method where injected nodes are connected to the nodes in the graph ? The limitation of previous works claimed by the authors seem rather arbitrary and inconsistent. If the ability to modify the graph structure is not a realistic assumption, then connecting the injected nodes to the real nodes in the graph should similarly be considered unrealistic. Basically, if connecting injected nodes to the real nodes in the graph is permissible, then it should be permissible for previous fairness attack methods to modify the graph structure by adding edges to it (without deleting existing edges).

- The utilized 1% perturbation rate is rather high, in real-world large graph datasets consisting of millions of nodes, this is equivalent to injecting the graph with tens of thousands of nodes which can hardly be considered unnoticeable. The authors should experiment with significantly smaller perturbation rates and compare the corresponding results against the relevant baselines in the literature.

- It is not possible to evaluate the effectiveness of proposed method against the relevant fairness attack methods using a single dataset only as 2 out of the 3 fairness attack methods are reported without results on 2 datasets. The authors should include additional datasets and/or additional Fairness attack methods.

- All 3 fairness GNN Attack methods [11, 13, 40] report results on one or both Pokec datasets. Therefore, if the author do not have access to the computational resources required to run the aforementioned baselines, they should run their proposed method on the setups of those 3 baselines and report the results of the 3 baselines  [11, 13, 40] from the corresponding works. In this manner, we would be able to properly evaluate the effectiveness of the proposed method against the relevant Fairness Attack baselines in the literature.

- It is unclear how the authors ensure a fair budget across all attack methods given the different nature of the attacks where some methods modify the graph structure and node features, while other methods inject the graph with additional nodes. Could the authors please elaborate on this point and how they ensured fairness across budgets of different types of attack methods ?

- The train/val/test split utilized assigns half of the nodes to labeled training nodes. However, in most realistic scenarios a significantly smaller percentage of the nodes are assigned to labeled training nodes. How does the proposed method perform when the number of training nodes in the graph is limited ? This is specially important given that targeted nodes with high uncertainty in this work are a subset of the training nodes. The authors should conduct experiments with limited labeled nodes and evaluate the effectiveness of the proposed method under this more realistic scenario.

- The authors should evaluate their proposed method on additional common benchmark datasets for Fair GNN learning task such as Credit, Bail and NBA datasets.

**Questions:**

Please refer to the Weaknesses section.

**Limitations:**

The authors adequately discuss the limitations of their proposed method.

---

> ### Author Rebuttal · Authors · 2024-08-05
>
> We would like to thank the reviewer for these insightful comments. Specifically, we aim to address the concerns of the reviewer with the following responses.
>
> ------
>
> **[Concerns about the attack scenarios (W1, W2)]**
>
> Take a social graph like Twitter as an example, where each node denotes a user and each link represents a following relationship. In this case, modifying the edges between existing nodes means alternating the following relationship between real users. **Such operations usually ask the attackers to hack the user accounts, which is hard and time-consuming.** However, with the node-injection-based attack, the attackers only need to **create several zombie accounts (node-injection) and follow several real users**, which is much easier.
>
> In fact, compared with attacks modifying existing edges, **the superiority of node-injection-based attacks has been widely verified in multiple prior works [1-4]**. However, all these works only focus on attacking GNNs utility, while neglecting the fairness vulnerability of GNNs.
>
> ------
>
> **[Concerns about the perturbation ratio (W3)]**
>
> 1. It seems that there are some misunderstanding about the perturbation ratio. In our work, the perturbation ratio **is based on the labeled nodes** in the graph instead of all nodes in the graph. Specifically, as we introduced in Appendix G, the injected nodes for Pokec-z, Poker-n and DBLP are only 102, 87 and 32 respectively, which is around **0.15%**, **0.13%** and **0.16%** of all nodes in the original graph.
>
> 2. In fact, the proportion of node injections in our work is comparable to or even lower than that of other related node-injection-based attacks. To support our claim, we statistics the default node injection ratios (relative to the total number of nodes in the graph) of several node-injection-based attacks below:
>
>    |                           | NIPA[1] |  TDGIA[2]   | MaxiMal[4] |    Ours     |
>    | :-----------------------: | :-----: | :---------: | :--------: | :---------: |
>    | **Node injection ratios** |   1%    | 0.07%-0.30% |     1%     | 0.13%-0.16% |
>
> 3. **We also conduct experiments by decreasing the perturbation ratio to 0.08%** (relative to the total number of nodes) on DBLP, i.e. 16 injected nodes. The attack performance on GCN is shown **in Table R2 in the uploaded PDF.** It can be seen that, even with a more limited perturbation rate, NIFA still achieves the best fairness attack performance compared with other baselines.
>
> ------
>
> **[Concerns about the baselines & datasets (W4, W5, W8)]**
>
> - **Baselines:** Thanks for your constructive feedback. To the best of our knowledge, **FA-GNN, FATE and G-FairAttack are the only three attack methods on GNN fairness.** We would be more than happy to provide additional comparisons if the reviewer can clarify other missing baselines on GNN fairness attacks.
>
> - **Datasets:** In fact, our datasets are consistent with FA-GNN, the first fairness attacks on GNNs. We agree that comparisons on more datasets could better verify the effectiveness of NIFA. In detail, we further conduct experiments on the setup of FATE and G-FairAttack, i.e. Pokec datasets with fewer nodes. We also examine the effectiveness of NIFA on the German benchmark, which is widely used in previous GNN fairness studies [5-7]. The experimental results are shown **in Table R4 in the uploaded PDF**, where NIFA still achieves competitive fairness attack performance.
>
> ------
>
> **[Concerns about the fair budget (W6)]**
>
> For methods that only inject new nodes, such as AFGSM, TDGIA and G2A2C, we require the number of injected nodes and average degree of injected nodes to be the same as ours. For methods that require to modify the graph structure, such as FA-GNN, FATE and G-FairAttack, **we set the number of modified edges to be the same with our injected edges**, i.e. the added edges between injected nodes and original nodes. We will enhance the clarity of this part in Appendix G in the future.
>
> ------
>
> **[Concerns about the label ratio (W7)]**
>
> Thanks for your comments! In fact, our datasets are collected from the previous work -- FA-GNN, and our train/val/test ratios are set to be consistent with its settings.
>
> We agree that there might be fewer labeled nodes in more realistic scenarios. To evaluate the effectiveness of NIFA under such settings, we decrease the training ratio from 50% to 25%, 10% and 5%, respectively. The attack performance of NIFA is shown **in Table R3 in the uploaded PDF**. It can be seen that, even with much fewer labeled training nodes, NIFA still consistently demonstrates promising attack performance.
>
> ------
>
> We sincerely thank the reviewer for the thoughtful comments and constructive feedback. We sincerely hope that our responses can effectively address your concerns and contribute to a better version of our research. If you have any further questions or confusion, please do not hesitate to reach out to us. We would be more than willing to assist and provide further clarification.
>
> ------
> **References**
>
> [1] Adversarial Attacks on Graph Neural Networks via Node Injections: A Hierarchical Reinforcement Learning Approach, WWW 2020.
>
> [2] TDGIA: Effective Injection Attacks on Graph Neural Networks, KDD 2021.
>
> [3] Let Graph be the Go Board: Gradient-free Node Injection Attack for Graph Neural Networks via Reinforcement Learning, AAAI 2023.
>
> [4] Maximizing Malicious Influence in Node Injection Attack, WSDM 2024.
>
> [5] EDITS: Modeling and Mitigating Data Bias for Graph Neural Networks, WWW 2022.
>
> [6] Improving Fairness in Graph Neural Networks via Mitigating Sensitive Attribute Leakage, KDD 2022.
>
> [7] FairSIN: Achieving Fairness in Graph Neural Networks through Sensitive Information Neutralization, AAAI 2024.

---

> > ### Comment · Reviewer_U6JX · 2024-08-12
> >
> > Thank you to the authors for their response. After carefully reviewing it, I have updated my original score.

---

> ### Author Response · Authors · 2024-08-12
> **Thank you for the reply**
>
> Thanks for your valuable feedback and reconsideration of scores!

---

### Official Review · Reviewer_L3XD · 2024-07-11

**Soundness:** 3
**Presentation:** 4
**Contribution:** 3
**Rating:** 7
**Confidence:** 4

**Summary:**

This paper examines the vulnerability of GNN fairness under adversarial attacks.  A gray-box node injection-based poisoning attack method, namely NIFA, is proposed. NIFA follows the newly designed uncertainty maximization principle and homophily-increase principle. Then,  multiple novel objective functions are proposed to guide the optimization of the injected nodes’ features, impacting the victim GNN’s fairness from a feature perspective. The experiment is extensive and solid.

**Strengths:**

S1: The problem of the vulnerability of GNN fairness is interesting and very important. This paper is well-motivated.

S2: The proposed method is technically sound.

S3: The experiment is solid and extensive. Very comprehensive experimental results are reported in the appendix including hyper-parameter testing, ablation studies, analysis of poisoned graph, etc.

S4: The paper is well-written and very easy to follow.

**Weaknesses:**

W1: Node injection will change the topological structure of a graph. However, several existing studies work on structural fairness in GNN. I am wondering whether the proposed fairness attacks are applicable to those works. What if the graph structure changes over time and becomes dynamic? Is the proposed attack applicable to dynamic fairness GNN methods?

Some references:

[1] Uncovering the Structural Fairness in Graph Contrastive Learning. NeurPIS 2022.

[2] Tail-GNN: Tail-Node Graph Neural Networks. KDD. 2021.

[3] On Generalized Degree Fairness in Graph Neural Networks. AAAI. 2023.

[4] Toward Structure Fairness in Dynamic Graph Embedding: A Trend-aware Dual Debiasing Approach. KDD. 2024.

**Questions:**

Please see the weakness.

**Limitations:**

The authors have adequately addressed the limitations.

---

> ### Author Rebuttal · Authors · 2024-08-05
>
> We would like to thank the reviewer for these insightful comments. Specifically, we aim to address the concerns of the reviewer with the following responses.
>
> ------
>
> **[Question about the Structural Fairness (W1)]**
>
> Thanks for your inspiring question! According to prior research, structural fairness and attribute fairness stem from different reasons [3]. The structural fairness (such as degree fairness in TailGNN [2] and DegFairGNN [3]) may be caused by the limited neighborhood information, while attribute fairness mainly comes from the "homophily principle" in GNN message propagation and the correlation between sensitive attributes and other attributes. In this way, the rationale behind NIFA such as the "homophily-increase principle" may be more suitable for attribute fairness instead of structural fairness.
>
> However, we highly agree that the adversarial attacks on structural fairness would be another inspiring research direction in the future, and we would like to add this to our discussion in the next version.
>
> ------
>
> **[Question about the Dynamic Fairness (W1)]**
>
> To the best of our knowledge, the dynamic fairness in GNN is still under-explored, and the only related work is based on structural fairness [1], which is not suitable for NIFA as we discussed before. We extend our best gratitude for your insightful feedback, and we believe that more efforts are still in demand for a more detailed definition of dynamic fairness based on the sensitive attributes before launching a corresponding fairness attack.
>
> ------
>
> We sincerely thank the reviewer for the encouraging comments and thoughtful feedback. We sincerely hope that our responses can effectively address your concerns and contribute to a better version of our research. If you have any further questions or confusion, please do not hesitate to reach out to us. We would be more than willing to assist and provide further clarification.
>
> **Reference**
>
> [1] Toward Structure Fairness in Dynamic Graph Embedding: A Trend-aware Dual Debiasing Approach. KDD. 2024.
>
> [2] Tail-GNN: Tail-Node Graph Neural Networks. KDD. 2021.
>
> [3] On Generalized Degree Fairness in Graph Neural Networks. AAAI. 2023.

---

> > ### Comment · Reviewer_L3XD · 2024-08-13
> >
> > Thank you very much for your responses which address my concerns well. I will keep my score.

---

### Official Review · Reviewer_ZRPp · 2024-07-12

**Soundness:** 3
**Presentation:** 3
**Contribution:** 2
**Rating:** 6
**Confidence:** 4

**Summary:**

This paper proposes NIFA, a novel fairness attack method via node injection. In particular, the authors use the uncertainty maximization principle to select the target node to attack and randomly connect the injected nodes to the target nodes in the same sensitive group to increase the overall homophily. Finally, the authors use direct optimization to tune the features of the injected nodes to maximize the unfairness and minimize the predictive loss. Experimental results show the effectiveness of NIFA in reducing the fairness of GNNs.

**Strengths:**

- This paper delivers the first node injection attacks on fairness for GNNs.
- The paper is overall well-organized and easy to follow.
- The methodology is basically well-motivated and verified by empirical studies.

**Weaknesses:**

- Although experiments show that NIFA distinctly increases the bias level of GNNs, the accuracy decreases as well, especially for DBLP. It can be helpful if more discussions on the tradeoff between accuracy and bias level are provided.
- Complexity analysis is not included in the paper. A succinct discussion (theoretical or empirical) on the computational efficiency of NIFA is encouraged.
- There exists a gap between Lemma 1 and the goal of the homophily-increase principle. Increased homophily cannot guarantee the increase of $\Delta_{sp}$ and $\Delta_{eo}$.
- The overall framework of NIFA is simple but kind of incremental. The technical novelty and provided insights are somewhat limited.

**Questions:**

Please see the weaknesses.

**Limitations:**

The limitations including the potential negative societal impact of this work are sufficiently discussed in the paper.

---

> ### Author Rebuttal · Authors · 2024-08-05
>
> We sincerely thank the reviewer for the constructive comments, and we aim to address the concerns with the following responses.
>
> ------
>
> **[Concerns about the utility (W1)]**
>
> We will first analyze the potential reasons for utility decrease, and then provide some solutions for balancing the trade-off between utility and fairness attack.
>
> - **Potential reasons:** Compared with evasion attacks, where only input data is modified and the victim model remains unchanged, poisoning attacks naturally result in a larger utility change due to the change of victim model.
> - **Mitigation solutions:** There are some potential methods to control the trade-off between utility and fairness of NIFA, **such as controlling the number of target nodes**, i.e. the nodes with the top $k$ highest uncertainty. Intuitively, with more nodes connected with injected nodes, the utility will be more likely to be influenced. To support our claims, we tune the $k$ in a range of {0.10, 0.25, 0.50, 0.75} in DBLP, and the attack performance on GCN and GraphSAGE is shown **in Table R1 in the uploaded PDF**. It can be seen that, after decreasing $k$, the utility after the attack can be effectively preserved.
>
> ------
>
> **[Concerns about the complexity (W2)]**
>
> We have included an efficiency analysis for NIFA **in Appendix H.5**. The results indicate that NIFA demonstrates much higher efficiency compared with other fairness attacks on GNNs.
>
> ------
>
> **[Concerns about the Lemma 1 (W3)]**
>
> As we claimed in footnote 2, the relationship between homophily ratio and fairness is widely discussed in previous work. For a better understanding of the role of homophily ratio in GNN fairness and to mitigate the gap, we provide the following theoretical analysis:
>
> - **Theoritical analysis:** Denote $P(s|x)$ as a predictor that estimates the sensitive attributes $s$ given node features $x$. Inspiring by [1], we utilize a linear intensity function $\mathcal{D} _{\theta}(s|x)$ with parameter $\theta$ to define its predictive capability, where $\mathcal{D} _{\theta}(s|x) \sim \mathcal{N}(\mu, \sigma^2)$ and $\mathcal{D} _{\theta}(\overline{s}|x) \sim \mathcal{N}(\overline{\mu}, \sigma^2)$, where $\overline{s}$ is the false sensitive attribute and $\mathcal{N}(\cdot, \cdot)$ is the Gausssian distribution. In this way, $\mu > \overline{\mu}$ indicates that $\mathcal{D} _{\theta}$ prefers to provide a higher intensity score for the true sensitive attribute given the node embeddings $x$, and the larger $\mu-\overline{\mu}$ is, the stronger inference capability of $\mathcal{D} _{\theta}$ and more biased node embeddings $x$ are.
>
>   ​    We further formulate the message propagation process in GNN as $x_i^{\prime}=x _i+x _i^{neigh}$, where $x _i^{neigh}$ denotes the average neighbor embeddings for node $i$, and $x _i^{neigh}=P _i^{same}x _i^{same}+P _i^{diff}x _i^{diff}$. $P _i^{same}$ denotes the probability of selecting a neighbor with the same sensitive attribute with node $i$, i.e. homophily ratio, and $P _i^{same} + P _i^{diff}=1$. $x _i^{same}$ denotes the average neighbor embeddings with the same sensitive attribute with node $i$.
>
>   ​    In this way, we can have the following derivation of the equations:
>   $$
>   \\begin{align*}
>   \\mathbb{E}\\left\\{ \\mathcal{D} _{\\theta}(s _i \\mid x _i') - \\mathcal{D} _{\\theta}(\\overline{s _i} \\mid x _i') \\right\\}
>   &= \\mathbb{E}\\left\\{ \\mathcal{D} _{\\theta}(s _i \\mid x _i) + \\mathcal{D} _{\\theta}(s _i \\mid x _i^{\\text{neigh}}) - \\mathcal{D} _{\\theta}(\\overline{s _i} \\mid x _i) - \\mathcal{D} _{\\theta}(\\overline{s _i} \\mid x _i^{\\text{neigh}}) \\right\\} \\\\
>   &= \\mathbb{E}\\left\\{ \\mathcal{D} _{\\theta}(s _i \\mid x _i) - \\mathcal{D} _{\\theta}(\\overline{s _i} \\mid x _i) + P_i^{\\text{same}} \\mathbb{E}\\left\\{ \\mathcal{D} _{\\theta}(s _i \\mid x _i^{\\text{same}}) - \\mathcal{D} _{\\theta}(\\overline{s _i} \\mid x _i^{\\text{same}}) \\right\\} \\right. \\\\
>   &\\quad \\left. - P _i^{\\text{diff}} \\mathbb{E}\\left\\{ \\mathcal{D} _{\\theta}(s _i \\mid x _i^{\\text{diff}}) - \\mathcal{D} _{\\theta}(\\overline{s _i} \\mid x _i^{\\text{diff}}) \\right\\} \\right\\} \\\\
>   &= \\mathbb{E}\\left\\{ \\mathcal{D} _{\\theta}(s _i \\mid x _i) - \\mathcal{D} _{\\theta}(\\overline{s _i} \\mid x _i) + \\left( P _i^{\\text{same}} - P _i^{\\text{diff}} \\right) (\\mu - \\overline{\\mu}) \\right\\} \\\\
>   \\end{align*}
>   $$
>
>   ​    It can be seen that, the second term $\\left( P _i^{\\text{same}} - P _i^{\\text{diff}} \\right) (\\mu - \\overline{\\mu})$  $= ( 2*P _i^{\\text{same}}- 1) (\\mu - \\overline{\\mu})$ **is positively correlated to the homophily ratio**, and with the increase of homophily ratio as we introduced in Lemma1, the message propagation process will result in larger unfairness.
>
> ------
>
> **[Concerns about the contributions (W4)]**
>
> The major contributions of this paper are three-fold:
>
> 1. We are the first to launch a node-injection-based fairness attack on GNNs to the best of our knowledge, and highlight the vulnerability of GNN fairness.
> 2. We design a simple yet effective method NIFA, which consists of two novel and insightful principles for guiding node injection operations. Extensive experiments verify the effectiveness, deceptiveness and efficiency of NIFA, which can effectively deteriorate GNN fairness, even including fair GNNs with merely 1% injected nodes.
> 3. From the perspective of responsible AI, we summarize several key insights from the success of NIFA, which we believe can inspire more in-depth research on robust GNN fairness.
>
> ------
>
> We sincerely hope that our responses can effectively address your concerns and contribute to a better version of our research. If you have any further questions or confusion, please do not hesitate to reach out to us. We would be more than willing to assist and provide further clarification.
>
> **Reference**
>
> [1] FairSIN: Achieving Fairness in Graph Neural Networks through Sensitive Information Neutralization, AAAI 2024.

---

> > ### Comment · Reviewer_ZRPp · 2024-08-12
> >
> > I thank the authors for providing a detailed response. The provided analysis of the connection between homophily ratio and bias seems to rely on strong assumptions, but it would be fine if this has been verified empirically by previous works. Most of my concerns are well addressed. Hence, I am happy to raise my score.

---

> > > ### Author Response · Authors · 2024-08-12
> > >
> > > Thanks again for your valuable comments and reconsideration of scores! To better understand the relationship between the homophily ratio and fairness, we would like to provide a more thorough discussion and summarization of previous related work in the final version.

---

### Author Rebuttal · Authors · 2024-08-05

## To all reviewers:

We sincerely thank all reviewers for their valuable feedback and encouraging comments on our paper. All four reviewers consistently approve of the topic: **“interesting”**, **“well-motivated”**, and **“encouraging”**, and the presentation of our work: **"well-written"**, **"very easy to follow"** and **"highly readable"**. Also, most of the reviewers agree that our experimental results:**“significant effectiveness”**, **“solid and extensive”**, and **“comprehensive”**.

We also notice that there are some misunderstanding and concerns about our paper. **Due to space limitations, we have included most of the experimental results (figures and tables) in the attached PDF, with corresponding references provided in the rebuttal.** We sincerely hope that our responses have effectively addressed your concerns and contributed to a deeper understanding of our research. If you have additional questions or confusion, please feel free to contact us without hesitation. We would sincerely like to provide more information and clarification if necessary.

------

## To reviewer #wXHX:

Due to space limitations, we place the theoretical analysis for the relationship between homophily ratio and fairness here.

**Theoritical analysis:** Denote $P(s|x)$ as a predictor that estimates the sensitive attributes $s$ given node features $x$. Inspiring by [1], we utilize a linear intensity function $\mathcal{D} _{\theta}(s|x)$ with parameter $\theta$ to define its predictive capability, where $\mathcal{D} _{\theta}(s|x) \sim \mathcal{N}(\mu, \sigma^2)$ and $\mathcal{D} _{\theta}(\overline{s}|x) \sim \mathcal{N}(\overline{\mu}, \sigma^2)$, where $\overline{s}$ is the false sensitive attribute and $\mathcal{N}(\cdot, \cdot)$ is the Gausssian distribution. In this way, $\mu > \overline{\mu}$ indicates that $\mathcal{D} _{\theta}$ prefers to provide a higher intensity score for the true sensitive attribute given the node embeddings $x$, and the larger $\mu-\overline{\mu}$ is, the stronger inference capability of $\mathcal{D} _{\theta}$ and more biased node embeddings $x$ are.

We further formulate the message propagation process in GNN as $x_i^{\prime}=x _i+x _i^{neigh}$, where $x _i^{neigh}$ denotes the average neighbor embeddings for node $i$, and $x _i^{neigh}=P _i^{same}x _i^{same}+P _i^{diff}x _i^{diff}$. $P _i^{same}$ denotes the probability of selecting a neighbor with the same sensitive attribute with node $i$, i.e. homophily ratio, and $P _i^{same} + P _i^{diff}=1$. $x _i^{same}$ denotes the average neighbor embeddings with the same sensitive attribute with node $i$.

In this way, we can have the following derivation of the equations:
$$
\\begin{align*}
\\mathbb{E}\\left\\{ \\mathcal{D} _{\\theta}(s _i \\mid x _i') - \\mathcal{D} _{\\theta}(\\overline{s _i} \\mid x _i') \\right\\}
&= \\mathbb{E}\\left\\{ \\mathcal{D} _{\\theta}(s _i \\mid x _i) + \\mathcal{D} _{\\theta}(s _i \\mid x _i^{\\text{neigh}}) - \\mathcal{D} _{\\theta}(\\overline{s _i} \\mid x _i) - \\mathcal{D} _{\\theta}(\\overline{s _i} \\mid x _i^{\\text{neigh}}) \\right\\} \\\\
&= \\mathbb{E}\\left\\{ \\mathcal{D} _{\\theta}(s _i \\mid x _i) - \\mathcal{D} _{\\theta}(\\overline{s _i} \\mid x _i) + P_i^{\\text{same}} \\mathbb{E}\\left\\{ \\mathcal{D} _{\\theta}(s _i \\mid x _i^{\\text{same}}) - \\mathcal{D} _{\\theta}(\\overline{s _i} \\mid x _i^{\\text{same}}) \\right\\} \\right. \\\\
&\\quad \\left. - P _i^{\\text{diff}} \\mathbb{E}\\left\\{ \\mathcal{D} _{\\theta}(s _i \\mid x _i^{\\text{diff}}) - \\mathcal{D} _{\\theta}(\\overline{s _i} \\mid x _i^{\\text{diff}}) \\right\\} \\right\\} \\\\
&= \\mathbb{E}\\left\\{ \\mathcal{D} _{\\theta}(s _i \\mid x _i) - \\mathcal{D} _{\\theta}(\\overline{s _i} \\mid x _i) + \\left( P _i^{\\text{same}} - P _i^{\\text{diff}} \\right) (\\mu - \\overline{\\mu}) \\right\\} \\\\
\\end{align*}
$$

It can be seen that, the second term $\\left( P _i^{\\text{same}} - P _i^{\\text{diff}} \\right) (\\mu - \\overline{\\mu})$  $= ( 2*P _i^{\\text{same}}- 1) (\\mu - \\overline{\\mu})$ is positively correlated to the homophily ratio, and with the increase of homophily ratio as we introduced in Lemma1, the message propagation process will result in larger unfairness.

**Reference**

[1] FairSIN: Achieving Fairness in Graph Neural Networks through Sensitive Information Neutralization, AAAI 2024.

---

### Comment · Area_Chair_ThxB · 2024-08-11
**Rebuttal**

Dear Reviewers,

I kindly ask you to review the authors' rebuttals and provide feedback on whether they have adequately addressed your concerns or if further clarifications are needed. Please note that the discussion deadline is fast approaching on August 13th.

Thank you for your attention to this matter.

Best regards,
AC

---

### Decision · Program_Chairs · 2024-09-25

**Decision:**

Accept (poster)

**Comment:**

This paper proposes NIFA, a Node Injection-based Fairness Attack that reveals vulnerabilities in Graph Neural Networks (GNNs) by injecting new nodes instead of altering existing connections. NIFA utilizes principles of uncertainty maximization and homophily increase, combined with optimized node features, to effectively compromise GNN fairness. Experiments on real-world datasets show that NIFA can significantly undermine the fairness of GNNs, including fairness-aware models, with the injection of just 1% of nodes. Reviewers generally agreed that the problem of using node injection to compromise GNN fairness is compelling, and the proposed method has clear merits. In their rebuttal, the authors addressed most of the reviewers' concerns and provided additional experiments. I suggest the authors incorporate these changes into the final version if the paper is accepted at NeurIPS.